# Ceramide sensing by human SPT-ORMDL complex for establishing sphingolipid homeostasis

Tian Xie [1,3], Peng Liu [1,3], Xinyue Wu [1], Feitong Dong[1], Zike Zhang[1], Jian Yue[1], Usha Mahawar[2], Faheem Farooq[2], Hisham Vohra[2], Qi Fang[1], Wenchen Liu[1], Binks W. Wattenberg [2] ✉ & Xin Gong [1] ✉

The ORM/ORMDL family proteins function as regulatory subunits of the serine palmitoyltransferase (SPT) complex, which is the initiating and rate-limiting enzyme in sphingolipid biosynthesis. This complex is tightly regulated by cellular sphingolipid levels, but the sphingolipid sensing mechanism is unknown. Here we show that purified human SPT-ORMDL complexes are inhibited by the central sphingolipid metabolite ceramide. We have solved the cryo-EM structure of the SPT-ORMDL3 complex in a ceramide-bound state. Structure-guided mutational analyses reveal the essential function of this ceramide binding site for the suppression of SPT activity. Structural studies indicate that ceramide can induce and lock the N-terminus of ORMDL3 into an inhibitory conformation. Furthermore, we demonstrate that childhood amyotrophic lateral sclerosis (ALS) variants in the SPTLC1 subunit cause impaired ceramide sensing in the SPT-ORMDL3 mutants. Our work elucidates the molecular basis of ceramide sensing by the SPT-ORMDL complex for establishing sphingolipid homeostasis and indicates an important role of impaired ceramide sensing in disease development.

As sphingolipids play vital roles as structural membrane components and as both pro-survival and pro-apoptotic signaling molecules, their levels need to be strictly controlled for proper cellular function[1-5]. Dysregulation of sphingolipid metabolism has been implicated in various diseases[6-8]. Sphingolipid biosynthesis is a highly regulated process that is initiated by the serine palmitoyltransferase (SPT) complex in the endoplasmic reticulum[9,10]. In humans, the heterotrimeric SPT core enzymatic complex, composed of SPTLC1, SPTLC2 or SPTLC3, and SPTssa or SPTssb, catalyzes the first and rate-limiting step of de novo sphingolipid biosynthesis by condensation of serine and palmitoyl-CoA to generate 3-keto-dihydrosphingosine (3-KDS)[9-11]. 3-KDS is then rapidly reduced by 3-KDS reductase to form dihydrosphingosine[12], which is subsequently N-acylated by (dihydro) ceramide synthases with a second fatty acyl-CoA of variable acyl chain

lengths to produce dihydroceramide[13,14]. Dihydroceramide is further desaturated by dihydroceramide desaturase to yield ceramide[15], the precursor of most complex sphingolipids (Fig. 1a). Ceramide can be transformed into numerous sphingolipid metabolites, such as sphingosine, sphingosine-1-phosphate (S1P), ceramide-1-phosphate (C1P), sphingomyelin, and glycosphingolipids by a series of reversible enzymatic reactions[6]. Despite the enzymatic machineries responsible for sphingolipid synthesis having been uncovered during the last three decades, little is known regarding how those enzymes are regulated to preserve sphingolipid homeostasis at the molecular level.

The ORM/ORMDL proteins are a family of highly conserved ER membrane proteins found across kingdoms from yeast to mammals as well as in plants[16]. These proteins form conserved complexes with SPT and function as regulatory subunits of the SPT complex[1,2]. As a critical

[1]Department of Chemical Biology, School of Life Sciences, Southern University of Science and Technology, Shenzhen, Guangdong 518055, China.
[2]Department of Biochemistry and Molecular Biology, Virginia Commonwealth University School of Medicine, Richmond, VA 23298, USA. [3]These authors contributed equally: Tian Xie, Peng Liu. ✉e-mail: brian.wattenberg@vcuhealth.org; gongx@sustech.edu.cn

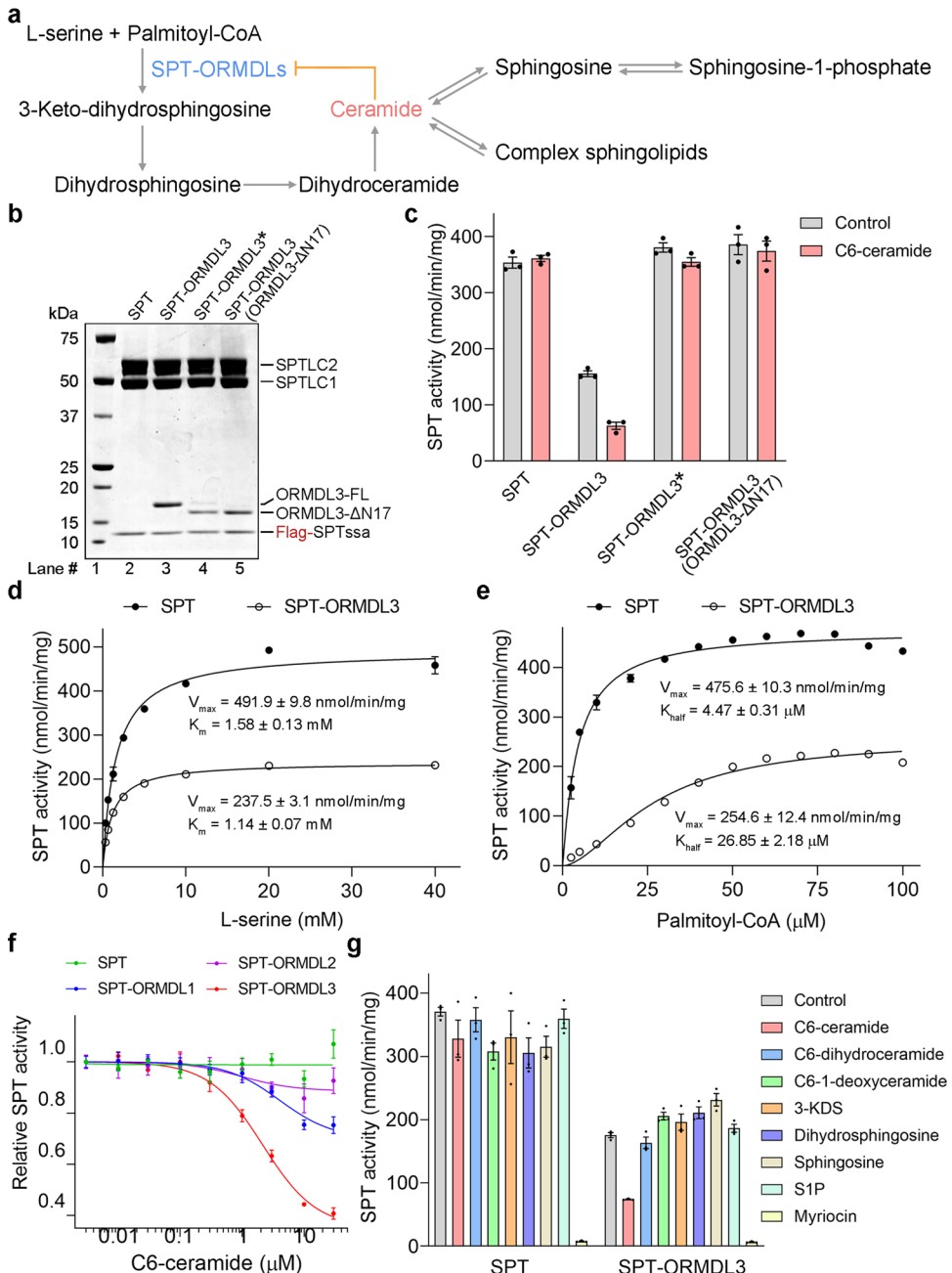

**Fig. 1 | Purified human SPT-ORMDL complexes can be inhibited by C6-ceramide. a** Overview of the de novo sphingolipid biosynthetic pathway and the homeostatic regulation of SPT activity by ceramide in an ORMDL-dependent manner. **b**, **c** Coomassie blue-stained SDS-PAGE gel (**b**) and the SPT activity (**c**) of purified SPT and various SPT-ORMDL3 complexes. ORMDL3* stands for the ORMDL3 construct from our previous study (ref. 27), which appears to express a mixture of minor full-length ORMDL3 protein (ORMDL3-FL) and dominant N-terminus truncated ORMDL3 protein. The newly made SPT-ORMDL3 (ORMDL3-FL) complex could be inhibited by C6-ceramide, whereas the SPT, SPT-ORMDL3*, and SPT-ORMDL3 (ORMDL3-ΔN17) complexes had no response to C6-ceramide. Data in (**c**) are presented as mean values ± SEM of three independent experiments. **d** SPT activity versus L-serine concentration measured using SPT or SPT-ORMDL3

complex. Data are presented as mean values ± SEM of three independent experiments. The data points were fitted with a Michaelis-Menten equation. **e** SPT activity versus palmitoyl-CoA concentration for SPT and SPT-ORMDL3 complex. Data are presented as mean values ± SEM of three independent experiments. The data points were fitted with an allosteric sigmoidal equation. **f** The inhibition curves of human SPT and SPT-ORMDL complexes by C6-ceramide in the in vitro SPT activity assay. Data are presented as mean values ± SEM of three independent experiments. All the estimated IC$_{50}$ values were summarized in Supplementary Table 1. **g** Selective inhibition of SPT-ORMDL3 by C6-ceramide among various sphingolipid species or analogs. Data are presented as mean values ± SEM of three independent experiments. The different sphingolipid species or analogs were applied at 10 μM in each assay. Source data are provided as a Source Data file.

mechanism for the homeostatic regulation of sphingolipid production, SPT activity is inhibited in response to elevated cellular sphingolipid levels in an ORM/ORMDL-dependent manner[3,5]. The yeast ORM proteins are phosphorylated at N-terminal serine sites upon reduced sphingolipid levels[1,17,18]. This phosphorylation releases SPT from ORM

inhibition, thus increasing sphingolipid biosynthesis and restoring sphingolipid levels. However, the mammalian ORMDL proteins lack the N-terminal phosphorylation sites, suggesting that the mammalian SPT-ORMDL complex might utilize a distinct mechanism to sense and respond to cellular sphingolipid levels[19]. Indeed, in isolated

membranes, the SPT-ORMDL complex is regulated by ceramide under conditions that preclude virtually all post-translational modifications[5]. The three human ORMDL isoforms (ORMDL1-3) share partially overlapping functions in the negative regulation of SPT activity[20,21]. Elevated ORMDL3 protein expression has a strong association with high susceptibility to asthma[22,23], although the precise mechanism remains unclear.

Exogenously added sphingosine has been shown to inhibit SPT activity via ORMDLs, and this inhibition could be blocked by the ceramide synthase inhibitor Fumonisin B1[3,24], suggesting that the added sphingosine must be converted to ceramide for SPT inhibition. Moreover, the ORM/ORMDL-dependent SPT inhibition by short-chain ceramide analogs has been demonstrated in intact/permeabilized cells and isolated membranes[3–5]. These results suggest that ceramide or a downstream sphingolipid metabolite can trigger the feedback inhibition of SPT activity via direct or indirect interaction with the SPT-ORM/ORMDL complex. However, elevated S1P levels caused by S1P lyase deficiency have also been suggested to inhibit SPT[25]. A recent study suggests that S1P is the key sphingolipid sensed by cells via S1PR/ORMDL axis to maintain sphingolipid homeostasis by a post-translational mechanism involving accelerated ORMDL turnover[26]. Further studies are required to establish the identity of sensed sphingolipid species and the corresponding sphingolipid sensor for homeostatic regulation of SPT activity. The recently reported cryo-EM structures of human SPT/SPT-ORMDL3 complexes by us and others uncover the principles of overall assembly and substrate selectivity, and yield clues about the regulation of SPT by ORMDLs[27,28]. However, the central question regarding how elevated sphingolipid levels lead to SPT inhibition through ORM/ORMDL remains unclarified. In this work, we demonstrate that ceramide binds directly with and inhibits the human SPT-ORMDL complexes, a process that is important for establishing cellular sphingolipid homeostasis.

## Results

### Purified human SPT-ORMDL complexes can be inhibited by C6-ceramide

In our previous study, the purified SPT and SPT-ORMDL3 complexes displayed similar enzymatic activity, and the added short-chain (and therefore soluble and amenable to addition to this system) ceramide analog (C6-ceramide) did not affect the activities of the complexes[27]. After careful examination of the ORMDL3 plasmid used in the previous study (hereafter named ORMDL3*), fourteen extra nucleotides were revealed upstream of the ORMDL3 open reading frame (ORF) (Supplementary Fig. 1a). By using a different restriction enzyme cutting site in the pCAG expression vector, we generated an alternative ORMDL3-FL expression plasmid, which lacks the extra 14 nucleotides upstream of the ORMDL3 ORF in the ORMDL3* plasmid (Supplementary Fig. 1b). Unexpectedly, the ORMDL3 protein expressed from the newly made ORMDL3-FL plasmid migrated as a single upper band on SDS-PAGE gel, while the ORMDL3 protein expressed from the ORMDL3* plasmid migrated as two bands, a minor upper band and a major lower band (Fig. 1b). Considering that the C-terminus of ORMDL3* is largely resolved in the previously solved SPT-ORMDL3* structure, whereas the N-terminus of ORMDL3* is flexible, we speculated that the minor upper band of the ORMDL3* protein might represent the full-length ORMDL3 (ORMDL3-FL) protein, while the major lower band of the ORMDL3* protein might represent an N-terminus truncated ORMDL3 protein. We also made an N-terminus truncated ORMDL3 construct (ORMDL3-ΔN17) that lacks the N-terminal 16 residue, which migrated similarly as the major lower band of the ORMDL3* protein on SDS-PAGE gel (Fig. 1b).

We observed that the specific activity of the newly made SPT-ORMDL3 complex at saturating conditions is around 50% of that of SPT complex (Fig. 1c–e). More importantly, the exogenously added C6-ceramide could inhibit the enzymatic activity of the newly made SPT-

ORMDL3 complex by approximately 60% with a half-maximal inhibitory concentration (IC$_{50}$) of 2.2 μM, but had no apparent effect on the SPT complex lacking ORMDLs (Fig. 1f, Supplementary Table 1). In contrast, the SPT-ORMDL3 (ORMDL3-ΔN17) complex displayed similar specific activity as the SPT and SPT-ORMDL3* complexes, and cannot be inhibited by exogenously added C6-ceramide (Fig. 1c). All together, these results suggest that the protein expressed from the ORMDL3* plasmid might be a mixture of the minor full-length ORMDL3 protein and the major N-terminus truncated ORMDL3 protein. A potential explanation for the expression of the truncation is outlined in Supplementary Fig. 1a. These data suggest that the N-terminus of ORMDL3 is critical for the regulation of SPT activity by ORMDL3 and ceramide.

We have also reconstituted the SPT-ORMDL1 and SPT-ORMDL2 complexes and demonstrated that C6-ceramide can only maximally inhibit SPT-ORMDL1 and SPT-ORMDL2 complexes by approximately 30% and 10%, respectively, with similar IC$_{50}$ values as that for the SPT-ORMDL3 complex (Fig. 1f, Supplementary Table 1). Our biochemical data demonstrate that C6-ceramide can be directly sensed by the SPT-ORMDL complexes to trigger ORMDL-dependent SPT suppression, and ORMDL3 is more responsive to C6-ceramide than ORMDL1 and ORMDL2.

To further investigate the specificity of the sphingolipid species sensed by SPT-ORMDL complexes, several different sphingolipid species or analogs were applied to the SPT and SPT-ORMDL complexes (Fig. 1g, Supplementary Fig. 2). Two close structural analogs of C6-ceramide, C6-dihydroceramide and C6-1-deoxyceramide, failed to induce the ORMDL3-dependent SPT inhibition at a concentration of 10 μM (Fig. 1g), suggesting that both the 4,5-trans-double bond and the C1-hydroxyl group in sphingosine backbone are important for C6-ceramide to inhibit SPT-ORMDL3. The ceramide precursors, 3-KDS and dihydrosphingosine, or the downstream sphingolipid metabolites, sphingosine and S1P, were also ineffective in inhibiting SPT-ORMDL3 (Fig. 1g). These results are consistent with the sphingolipid specificity reported earlier with membrane preparations, in which a strict ceramide stereospecificity was also observed[5]. These results indicate a direct and highly specific interaction between C6-ceramide and SPT-ORMDL3 complex. The SPT-specific inhibitor myriocin was included as a control for complete inhibition of all the SPT and SPT-ORMDL complexes (Fig. 1g, Supplementary Fig. 2b). The SPT-ORMDL1 and SPT-ORMDL2 complexes exhibited similar sphingolipid specificity as the SPT-ORMDL3 complex (Supplementary Fig. 2b), suggesting that the three ORMDL isoforms have minor differences in specificity for the sphingolipid species they can sense.

### Structure of the ceramide-bound SPT-ORMDL3 complex

To investigate the structural basis for SPT-ORMDL regulation by ceramide, we determined the structure of the SPT-ORMDL3 complex in the presence of excess C6-ceramide using single-particle cryo-EM (Supplementary Fig. 3). Through three-dimensional classification, two cryo-EM reconstructions of the dimeric SPT-ORMDL3 complex were obtained at overall resolutions of 3.3 Å and 3.2 Å, respectively, with differences in the relative orientations of the two monomeric protomers (Supplementary Fig. 3c). The structure of the monomeric SPT-ORMDL3 complex was resolved at an overall resolution of 2.9 Å (Supplementary Fig. 3c–f). The structural analysis hereafter is based on the monomeric complex because of its higher resolution.

Compared to the cryo-EM map of the monomeric SPT-ORMDL3* complex we previously reported[25], several extra well-ordered lipid-like densities could be resolved between ORMDL3 and SPTLC2 in the cytosolic membrane leaflet (Fig. 2a, b). While the N-terminus of ORMDL3* was not resolved in the SPT-ORMDL3* structure[27], the N-terminus of ORMDL3 could be well resolved in the updated maps (Fig. 2b). Apart from the differences in the N-terminal region of ORMDL3 and the presence of lipid-like densities between ORMDL3 and SPTLC2, the updated map and structure of the C6-ceramide-bound

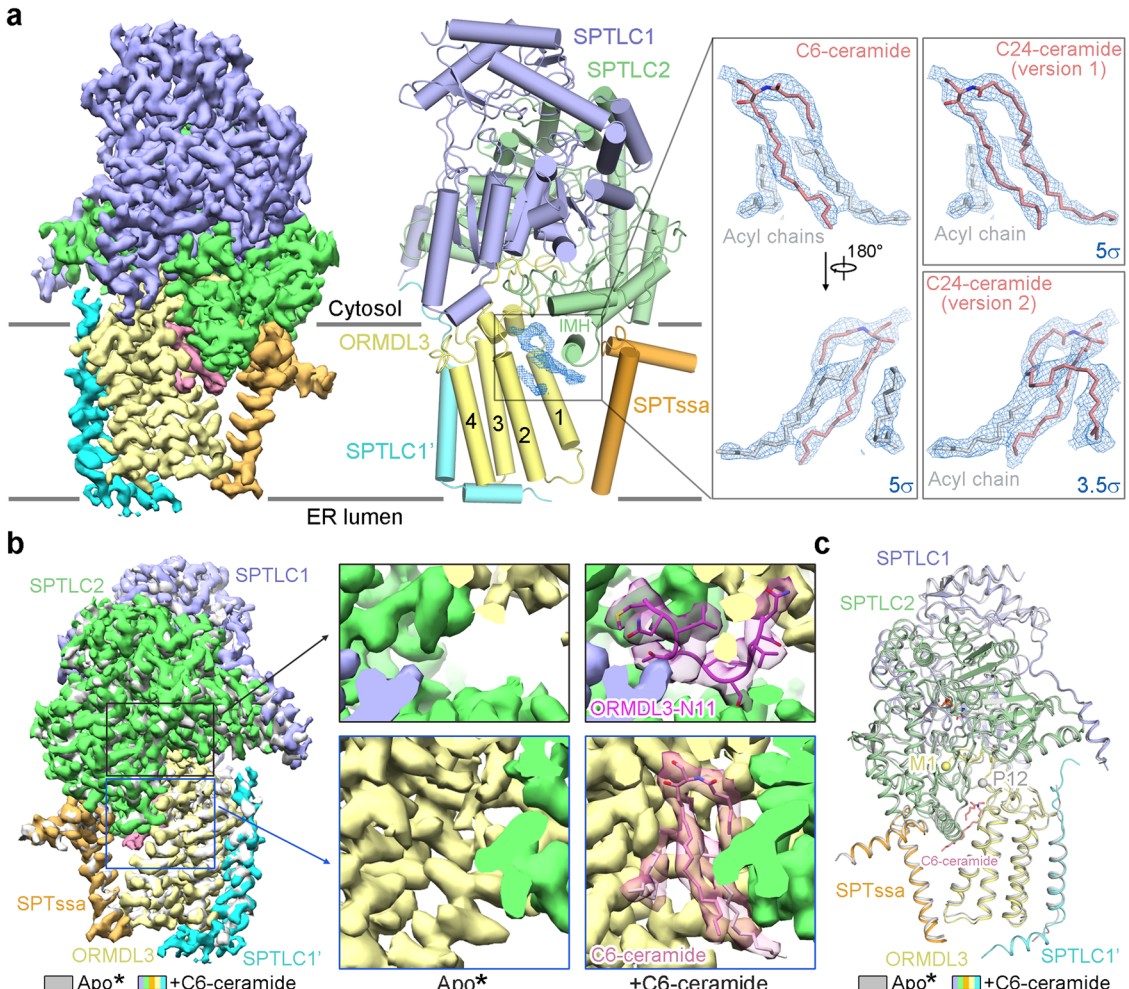

**Fig. 2 | Structure determination of the C6-ceramide-bound SPT-ORMDL3 complex. a** The cryo-EM map and overall structure of the monomeric C6-ceramide-bound SPT-ORMDL3 complex. SPTLC1 and SPTLC2 are shown in light blue and light green, respectively; SPTssa and ORMDL3 are shown in orange and light yellow, respectively; SPTLC1' is shown in cyan; the lipid-like density is shown in pink (left panel) or blue (right panels). Insets: The lipid-like density, shown in blue meshes, were contoured at 5σ or 3.5σ. C6-ceramide and C24-ceramide are shown as pink sticks; the acyl chains are shown as gray sticks. **b, c** Comparison of the EM maps (**b**) and structures (**c**) of the apo SPT-ORMDL3* complex and C6-ceramide-bound SPT-ORMDL3 complex. Apo* stands for the apo SPT-ORMDL3* complex used in our previous study[27]. The apo SPT-ORMDL3* complex (EMD-30080 and PDB 6M4O) is colored gray. The C6-ceramide-bound SPT-ORMDL3 complex is colored based on the subunits. The ORMDL3-N11 (the 11 residues at the N-terminus) and C6-ceramide in the C6-ceramide-bound SPT-ORMDL3 complex are shown as sticks, and the corresponding maps are colored semi-transparent magenta and pink, respectively (**b**, zoomed-in view, right). In contrast, the densities for the ORMDL3-N11 and C6-ceramide are not visible in the apo SPT-ORMDL3* complex (**b**, zoomed-in view, left). Met1 and Pro12, the resolved N-terminus of ORMDL3 and ORMDL3*, are shown as yellow and gray spheres, respectively (**c**).

SPT-ORMDL3 complex are very similar to those of the previous SPT-ORMDL3* complex (Fig. 2b, c).

Among the well-ordered lipid-like densities, the shape and size of one lipid-like density are consistent with a C6-ceramide molecule, showing one small polar head contiguous with two acyl chains (Fig. 2a). Close to the C6-ceramide density, we could observe two additional lipid-like densities, which might result from the acyl chains of phospholipids, sphingolipids, or other lipidic species. An alternative explanation for the two presumable acyl-chain densities is that they might arise from the average of C6-ceramide and endogenous bound long-chain ceramide (such as C24-ceramide) (Fig. 2a). Supporting this notion, we determined the structure of SPT-ORMDL3 complex in the absence of C6-ceramide (Supplementary Fig. 4) and observed a similar ceramide-like density in the apo SPT-ORMDL3 structure, which could be better fitted with C24-ceramide (Supplementary Fig. 4d). Except for the minor differences in the ceramide-like densities, the structures of the apo SPT-ORMDL3 and the C6-ceramide-bound SPT-ORMDL3 are nearly identical (Supplementary Fig. 5a, b). Ceramide molecules with

longer acyl chains could provide stronger hydrophobic interactions compared to short-chain C6-ceramide (Supplementary Fig. 5c). This is supported by previous data that demonstrated that as acyl-chain length increases ceramides becomes more potent at inhibiting SPT, as assayed in isolated membranes[5]. This could account for the observation that C6-ceramide only partially displaces endogenous long-chain ceramides, resulting in a mixture of C6-ceramide and natural long-chain ceramides in the C6-ceramide-bound SPT-ORMDL3 structure (Fig. 2a).

Furthermore, the natural long-chain ceramides (ranging from C14 to C24 in N-acyl chain length) could be identified from the purified SPT-ORMDL3 complex by both mass spectrometry and thin-layer chromatography (TLC) methods (Supplementary Fig. 5d, e). The endogenous bound ceramide in the structure of apo SPT-ORMDL3 could account for the ~50% reduction in the specific activity of SPT-ORMDL3 compared to that of SPT (Fig. 1c–e). Quantification from the TLC analysis indicates that the molar stoichiometry between the SPT-ORMDL3 protomer and bound ceramide is approximately 1:3 (Supplementary Fig. 5e). Considering that both phospholipids and

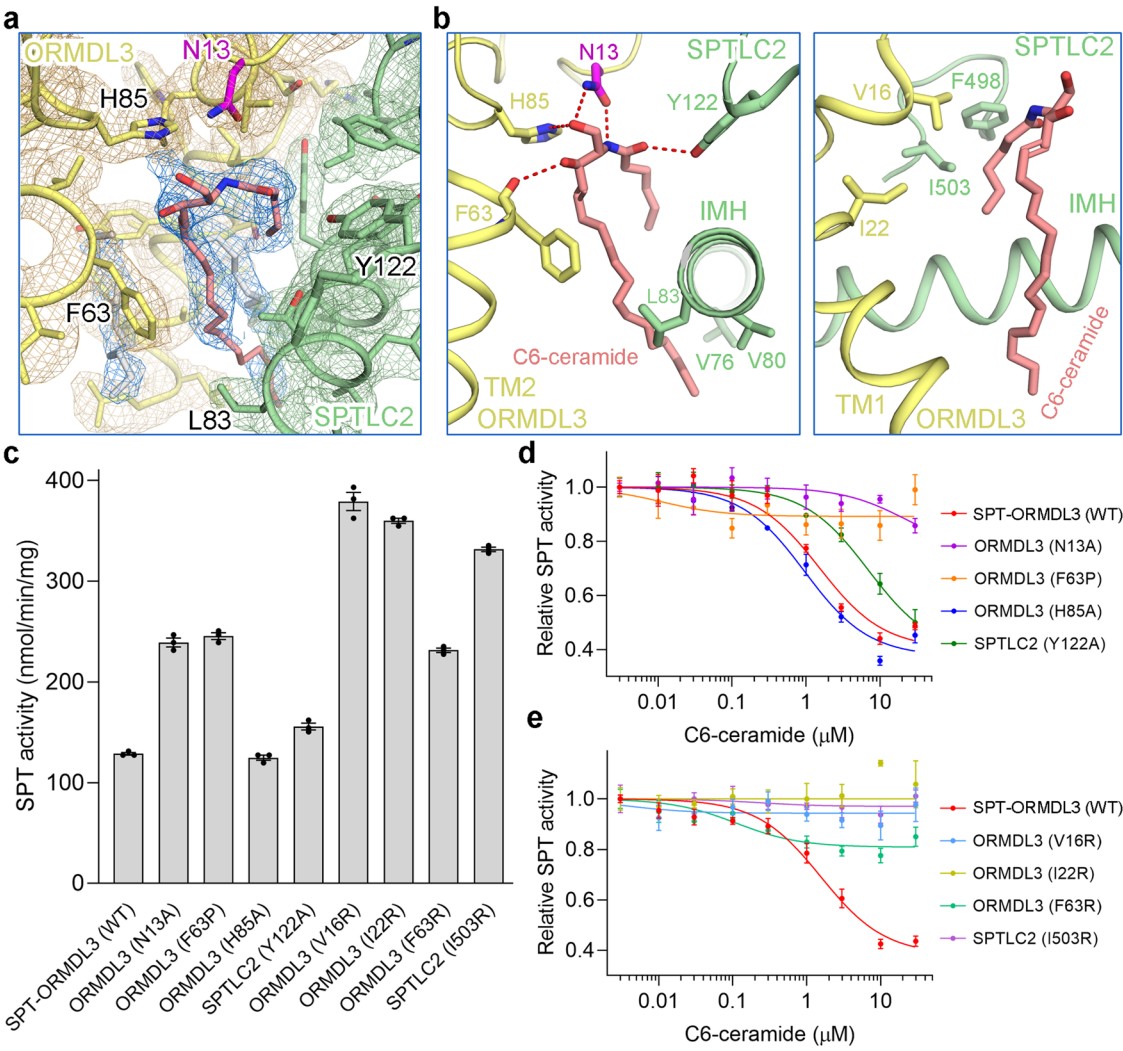

**Fig. 3 | The ceramide binding site in SPT-ORMDL3 complex. a** A close-up view of the density map for the ceramide binding site contoured at 6σ. **b** Detailed views of the ceramide binding site. Residues involved in the interactions are shown as sticks. All potential polar interactions are indicated by red dashed lines. **c** Functional characterization of the key residues involved in ceramide binding by SPT activity assay. Data are presented as mean values ± SEM of three independent experiments. **d, e** Inhibition curves of the SPT-ORMDL3 ceramide-binding variants by C6-ceramide. Curves for the variants mediating the polar interactions and hydrophobic interactions were presented in (**d**) and (**e**), respectively. Data are presented as mean values ± SEM of three independent experiments. Source data are provided as a Source Data file.

cholesterol were also detected in the isolated complex, it is likely that membrane lipids were bound in the detergent micelles associated with the isolated complex. Alternatively, there might be other ceramide binding sites in addition to the one observed in current structure. Both possibilities mentioned above may account for ceramide that remains binding when the presumptive ceramide binding residues were mutated, which will be described later.

**The ceramide binding site**

In the ceramide binding site (Fig. 3a), the polar head of ceramide makes five potential hydrogen bonds with the surrounding residues, including the side chain groups of Asn13 and His85 in ORMDL3, Tyr122 in SPTLC2, and the main chain group of Phe63 in ORMDL3 (Fig. 3b, left). To ascertain the function of these residues, we mutated the residues into alanine or proline. The SPT-ORMDL3 variants were individually purified to homogeneity and measured for the SPT activity (Supplementary Fig. 6a). The SPT-ORMDL3 variants containing ORMDL3-N13A or ORMDL3-F63P mutation displayed greatly enhanced enzymatic activity compared to the wild-type (WT) SPT-ORMDL3 complex (Fig. 3c). This result indicates that the ORMDL3-dependent

SPT inhibition has been largely abolished by the mutations, possibly owing to the disrupted or diminished ceramide binding. However, the SPT-ORMDL3 variants containing ORMDL3-H85A or SPTLC2-Y122A mutation exhibited similar activity as that of the WT SPT-ORMDL3 complex (Fig. 3c), suggesting a minor contribution of these two residues for ceramide binding. Consistent with the findings, the responses to C6-ceramide have been severely impaired for the former two variants, but largely sustained for the latter two variants (Fig. 3d).

The two aliphatic chains of ceramide are enclosed by numerous hydrophobic residues from both ORMDL3 and SPTLC2 (Fig. 3b). For the hydrophobic interface regarding the LCB chain of ceramide, only Phe63 of ORMDL3 was mutated to positively-charged arginine, as the other interface residues in IMH of SPTLC2 are buried in the hydrophobic membrane (Fig. 3b, left). For the hydrophobic interface considering the acyl-chain of ceramide, Val16 and Ile22 of ORMDL3, and Ile503 of SPTLC2 were chosen to be replaced by positively-charged arginine as they possess smaller side chains compared to Phe498 of SPTLC2 (Fig. 3b, right). All the mutations have limited effects on the association of ORMDL3 with SPT (Supplementary Fig. 6a). However, these mutations diminished the ORMDL3-dependent SPT inhibition

considerably (Fig. 3c), and disrupted the inhibition effect of C6-ceramide on these SPT-ORMDL3 variants rigorously (Fig. 3e). These results collectively support the identification of a functional ceramide binding site in the SPT-ORMDL3 complex that mediates the responsiveness of the complex to ceramide.

## The N-terminus of ORMDL3 is important for SPT inhibition

The N-terminus of ORMDL3 contacts with SPTLC1 and SPTLC2 through several polar interactions, and forms some intramolecular polar interactions within ORMDL3 (Fig. 4a, b). These polar interactions might contribute to the stabilization of the ORMDL3 N-terminus and lead to its well-resolved density in the updated cryo-EM maps. To determine the function of the ORMDL3 N-terminus, we generated three ORMDL3 N-terminal deletion variants, ORMDL3-ΔN2 (deletion of the Asn2 residue), ORMDL3-ΔN8 (deletion of residues 2–8), and the ORMDL3-ΔN17 mentioned in Fig. 1b (Supplementary Fig. 6b). These three SPT-ORMDL3 variants exhibited considerably improved enzymatic activity (Fig. 4c). More intriguingly, the inhibition effect of C6-ceramide on these SPT-ORMDL3 variants was nearly abolished (Fig. 4d). The Asn2 of ORMDL3 forms a hydrogen bond with the main chain of Leu330 in SPTLC1 (Fig. 4b). Mutation of Asn2 of ORMDL3 to alanine achieved effects similar to that of the ORMDL3 N-terminal deletion mutants (Fig. 4c, d), supporting the importance of the polar interaction between ORMDL3 N-terminus and SPTLC1. Taken together, these results establish that the ORMDL3 N-terminus is critical for the SPT inhibition by ORMDL3 and ceramide.

To investigate how ORMDL3 N-terminal mutations release SPT inhibition by ORMDL3 and disturb C6-ceramide-mediated SPT-ORMDL3 inhibition, we determined the structure of SPT-ORMDL3 variant bearing ORMDL3-ΔN2 mutation at an overall resolution of 3.1 Å for the monomeric complex (Supplementary Fig. 7). Although the overall structure of ORMDL3-ΔN2 mutant is closely identical to that of the WT complex (Supplementary Fig. 7d), the N-terminal segment of ORMDL3-ΔN2 is disordered (Fig. 4e), rather than inserted into the substrate binding site as it is in the WT structure. This supports a vital role of Asn2 in stabilizing the ORMDL3 N-terminus. The fixed N-terminus of ORMDL3 in the WT SPT-ORMDL3 complex presumably prevents the binding of palmitoyl-CoA to the enzyme (Fig. 4f). This is supported by the increased $K_{half}$ for palmitoyl-CoA in the ORMDL3-containing complex (Fig. 1e), whereas the flexible N-terminus of ORMDL3-ΔN2 would permit substrate binding, thus liberating SPT from ORMDL3 inhibition (Fig. 4c). In addition to the dissimilarities at the ORMDL3 N-terminus, the ceramide-like density sandwiched between ORMDL3 and SPTLC2 in the WT complex was largely diminished in the ORMDL3-ΔN2 mutant (Fig. 4e), suggesting that this mutation weakens ceramide binding. In line with this, the ORMDL3-ΔN2 mutation causes some local conformational changes for the residues near the ceramide binding pocket, especially Asn13 of ORMDL3, which would be expected to affect ceramide binding (Fig. 4g). Accordingly, the ceramide-enrichment from the purified ORMDL3-ΔN2 mutant was approximately 25% lower than that of the WT complex (Supplementary Fig. 5e, f). The remaining 75% may largely represent ceramide, along with other membrane lipids, bound to the detergent micelles in the isolated complex, or, some other potential ceramide binding sites in the complex. These results suggest that there is an important interplay between the N-terminus of ORMDL3 and the ceramide binding site.

## Ceramide locks the N-terminus of ORMDL3 into an inhibitory conformation

To elucidate the molecular defects of the ceramide binding mutant, we then determined the structure of the SPT-ORMDL3 variant containing ORMDL3-N13A mutation at an overall resolution of 2.9 Å for the monomeric complex (Supplementary Fig. 8). Structural comparison of this ceramide binding mutant with WT SPT-ORMDL3 reveals little conformational variations, except for two evident local structural

differences (Fig. 5a). Firstly, the ceramide-like density basically disappeared in the ORMDL3-N13A mutant (Fig. 5a), corroborating the largely disrupted ceramide binding by this point mutation. Consistent with this, the purified ORMDL3-N13A mutant demonstrated a decrease in ceramide enrichment of approximately 30% compared to the WT complex (Supplementary Fig. 5e, f). As with the ORMDL3-ΔN2 protomer, this demonstrates that the ORMDL3-N13A protomer is almost deficient in the ceramide binding site. Secondly, the N-terminal 10 residues of ORMDL3 became flexible in the ORMDL3-N13A mutant (Fig. 5a), explaining why this ceramide binding mutant is no longer able to suppress SPT activity efficiently (Fig. 3c, Supplementary Fig. 8d). Several residues that contributed to the intramolecular polar interactions at the ORMDL3 N-terminus in WT SPT-ORMDL3 structure, such as Asn11, Arg15, and Tyr91, exhibited apparent structural shifts in the ORMDL3-N13A mutant (Fig. 5b). We suggest that ceramide binding might trigger the conformational changes of Asn13 and the adjacent residues in ORMDL3, which could induce and lock the ORMDL3 N-terminus into an inhibitory conformation.

Although the SPT-ORMDL3 (ORMDL3-N13A) variant displayed a similar $V_{max}$ value to that of the WT SPT complex, the $K_{half}$ value of this variant toward palmitoyl-CoA was nearly two times as high as that of the WT SPT complex (Fig. 5c). This suggests that in the absence of ceramide binding, ORMDL3 can still inhibit SPT by reducing the affinity of SPT for palmitoyl-CoA substrate. Our results also suggest that the ceramide binding could stabilize the ORMDL3 N-terminus in an inhibitory conformation that blocks the palmitoyl-CoA binding, thus further repressing SPT-ORMDL3 complex via mixed inhibition that reduces the $V_{max}$ of SPT-ORMDL3 complex and further increases the $K_{half}$ toward palmitoyl-CoA (Fig. 5c). The SPT-ORMDL3 (ORMDL3-ΔN17) variant, lacking both the ORMDL3 N-terminus and the ceramide binding site, exhibited comparable $V_{max}$ and $K_{half}$ values as those of the WT SPT complex (Fig. 5c). The results indicate that the N-terminus of ORMDL3 is mostly accountable for the reduced affinity toward palmitoyl-CoA in the SPT-ORMDL3 complex.

## The impact of ORMDL3 mutations on the response of the SPT complex to ceramide in native membranes

The ORMDL3 mutations designed to test ceramide sensing and SPT inhibition by the ORMDL3 N-terminus described above were measured using purified proteins in detergent micelles. To assess the impact of those mutations in the native membrane environment, we capitalized on previous work demonstrating that ORMDL-dependent inhibition of SPT by ceramide can be reconstituted with isolated membranes[5]. To test the mutants in the absence of endogenous ORMDLs, we utilized a Hela cell line in which two of the three ORMDLs (ORMDL 2 and 3) had been deleted by Crispr/Cas9 gene editing, with the third ORMDL knocked down by siRNA transfection. The ORMDL3 construct to be tested was then co-expressed with SPT such that the complex was formed co-translationally (Supplementary Fig. 9a). To ensure stoichiometry of the SPT subunits, the SPT construct used was a fusion protein consisting of SPTLC1, SPTLC2, and SPTssa, which has previously been shown to respond to ceramide in an ORMDL-dependent manner identically to the endogenous SPT complex[4,5,29].

The ΔN17 mutation was severely impaired in its response to C8-ceramide, although there was a significant inhibition at high ceramide levels (Supplementary Fig. 9b). The ΔN8 mutation demonstrated a similar impairment (Supplementary Fig. 9c). These data confirm the importance of the ORMDL N-terminus. However, the response that remains after deleting these residues suggests that in the ceramide-induced inhibitory conformation, ORMDL3 mediates other structural changes in the SPT complex. This is addressed in more detail in the Discussion.

The deletion of Asn2 also blunts C8-ceramide inhibition (Supplementary Fig. 9d), albeit more subtly than when measured with purified protein. Similarly, the Asn13 to alanine mutant clearly impairs

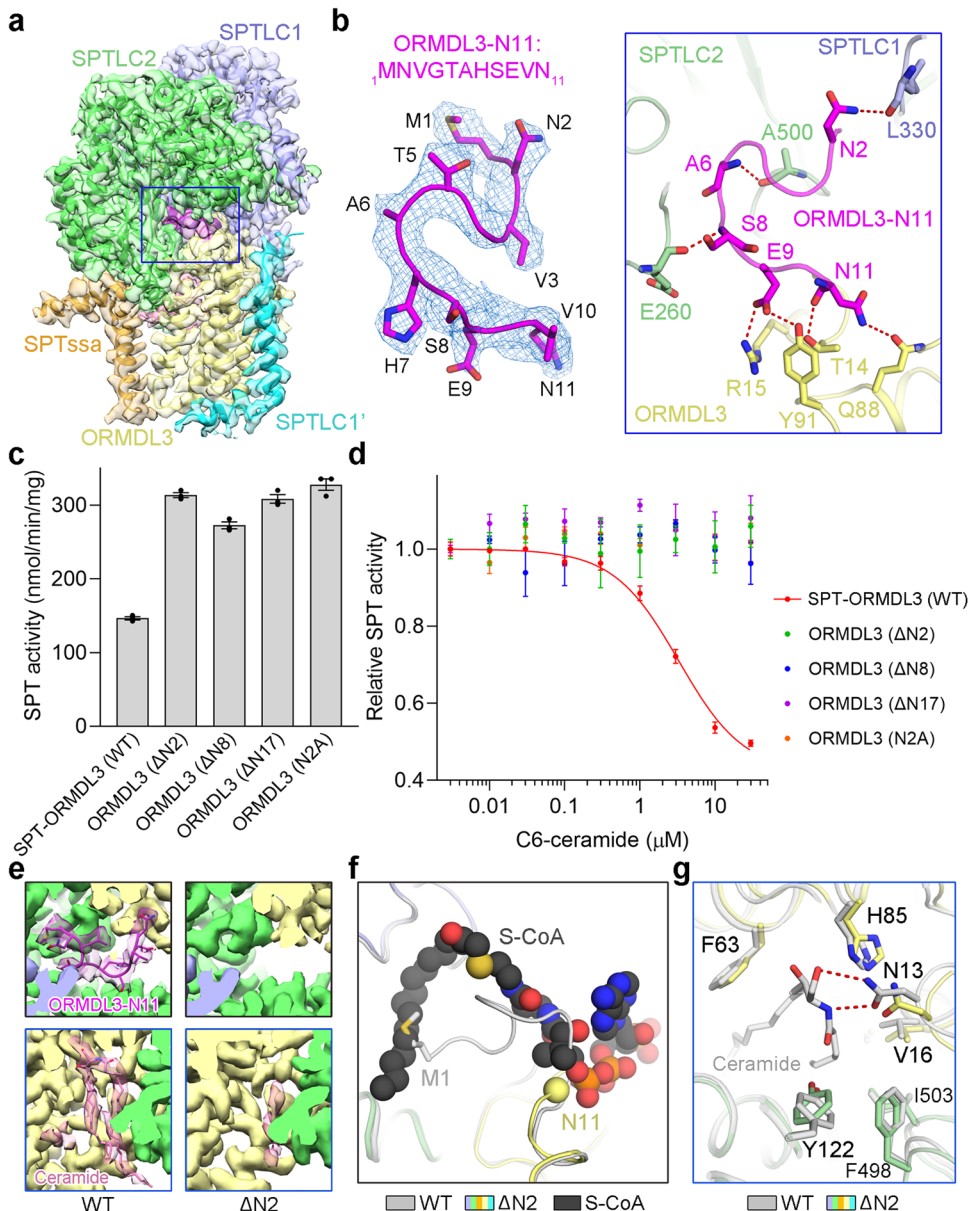

**Fig. 4 | The N-terminus of ORMDL3 is essential for SPT inhibition. a** The resolved N-terminus of ORMDL3 in the C6-ceramide-bound SPT-ORMDL3 map. The ORMDL3-N11 (the 11 residues at the N-terminus) are highlighted in magenta; all the other parts are shown in the same coloring scheme as above. The maps are shown as semi-transparent. **b** Close-up views of the density map and coordination of ORMDL3-N11. The electron densities for ORMDL3-N11, shown as blue mesh, were contoured at 6σ. The residues that form polar interactions with ORMDL3-N11 in SPTLC1, SPTLC2, and ORMDL3 are shown as sticks, and the potential polar interactions are indicated by red dashed lines. **c** Functional characterization of ORMDL3 N-terminal deletion variants and N2A variant by SPT activity assay. Data are presented as mean values ± SEM of three independent experiments. **d** The loss of C6-ceramide-mediated inhibition of SPT activity for the ORMDL3 N-terminal deletion variants and N2A variant. Data are presented as mean values ± SEM of three independent experiments. **e** Comparison of the EM maps for ORMDL3-N11 (upper) and ceramide (lower) in the apo wild-type (WT) SPT-ORMDL3 complex and the

ORMDL3-ΔN2 mutant. The ORMDL3-N11 is no longer visible in the ORMDL3-ΔN2 mutant. The ceramide-like density largely disappeared in the EM map of the ORMDL3-ΔN2 mutant. **f** Release of the N-terminus of ORMDL3 for substrate binding in the ORMDL3-ΔN2 mutant. The ORMDL3-ΔN2 mutant structure was superimposed with the C6-ceramide-bound WT SPT-ORMDL3 structure and S-CoA-bound SPT-ORMDL3* structure (PDB 7CQK). S-CoA, short for S-(2-oxoheptadecyl)-CoA and a nonreactive analog of palmitoyl-CoA, is displayed as black spheres. The C6-ceramide-bound SPT-ORMDL3 structure is colored gray, and the ORMDL3-ΔN2 mutant structure is colored based on the subunits as above. Asn11, the resolved N-terminus of ORMDL3-ΔN2, is shown as a yellow sphere. **g** Conformational changes of the ceramide-binding site in the ORMDL3-ΔN2 mutant. The ORMDL3-ΔN2 mutant structure (colored based on the subunits) was superimposed with the C6-ceramide-bound WT SPT-ORMDL3 structure (gray). Source data are provided as a Source Data file.

the ability of added C8-ceramide to inhibit SPT activity (Supplementary Fig. 9e), but to a lesser degree than when measured with purified protein. This quantitative difference may reflect a somewhat more ceramide-sensitive complex in the native membrane environment and/or an enhanced delivery of ceramide to the SPT/ORMDL ceramide binding site in native membranes.

## Impaired ceramide sensing by SPTLC1 variants associated with childhood ALS

A recent study has shown that the SPTLC1 variants, including Y23A, Δ39 (Leu39), and Δ40-41 (Phe40/Ser41), result in unrestrained sphingolipid biosynthesis and cause a monogenic form of childhood ALS[30]. These ALS-associated SPTLC1 variants locate near the transmembrane

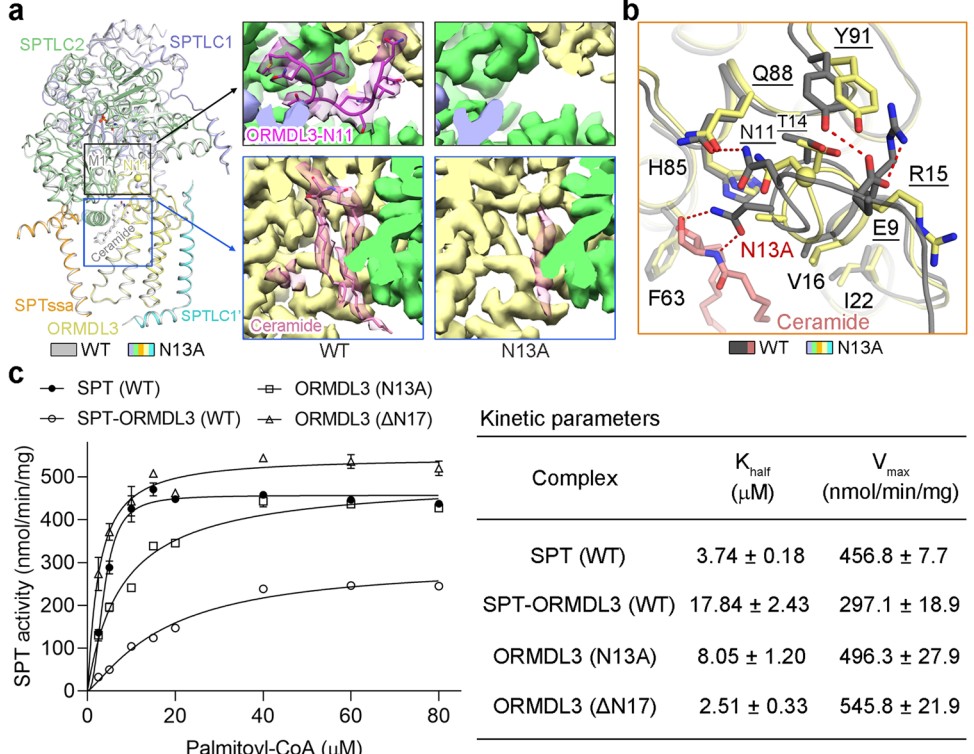

**Fig. 5 | Ceramide induces and locks the N-terminus of ORMDL3 into an inhibition conformation and the enzymatic kinetics of two SPT-ORMDL variants.** **a** Superposition of the apo WT SPT-ORMDL3 structure (gray) and the ORMDL3-N13A mutant structure (colored based on the subunits). Met1 and Asn11, the resolved N-terminus of WT ORMDL3 and ORMDL3-N13A, respectively, are shown as spheres. Insets: The EM maps for ORMDL3-N11 (upper) and ceramide-like density (lower) in the apo WT SPT-ORMDL3 and ORMDL3-N13A mutant structures. The ORMDL3-N11 is no longer visible in the ORMDL3-N13A mutant. The ceramide-like density largely disappeared in the EM map of the ORMDL3-N13A mutant.

**b** Conformational changes of the ceramide-binding site and the ORMDL3-N11 coordinating residues in the ORMDL3-N13A mutant structure. The ORMDL3-N13A mutant structure (colored based on the subunits) was superimposed with the C6-ceramide-bound WT SPT-ORMDL3 structure (gray). The residues involved in ORMDL3-N11 coordination are underscored. **c** Enzymatic kinetics of the ORMDL3-N13A and ORMDL3-ΔN17 variants compared to those of the WT SPT and SPT-ORMDL3 complexes. Data are presented as mean values ± SEM of three independent experiments. Source data are provided as a Source Data file.

interface between the N-terminal transmembrane helix (TM) of SPTLC1 and the TM3/4 of ORMDL3 (Fig. 6a). We suspected that the unrestrained sphingolipid biosynthesis caused by ALS-associated SPTLC1 variants might arise from impaired ceramide-mediated SPT-ORMDL inhibition. To test this, we generated the three SPTLC1 ALS variants and purified the three SPT-ORMDL3 variants to homogeneity (Supplementary Fig. 6c). The three SPTLC1 ALS variants exhibited little effect on ORMDL3 binding in the purified protein complexes (Fig. 6b), while displayed enhanced enzymatic activity compared to the WT SPT-ORMDL3 complex (Fig. 6c). The results indicate that the ORMDL3-dependent SPT suppression has been weakened by these SPTLC1 ALS variants. More importantly, the inhibition effect of C6-ceramide on all these SPTLC1 ALS variants was attenuated, especially for the SPTLC1 (Δ39) and SPTLC1 (Δ40-41) variants (Fig. 6d). Another study on the SPTLC1-ALS variants reported that ORMDLs had weakened interaction with the SPTLC1 (Δ39) and SPTLC1 (Δ40−41) variants as measured by co-immunoprecipitation and blue native PAGE, while the binding was retained with the SPTLC1 (Y23F) variant[31]. It is worth noting that the isolation of ORMDL with the other SPT complex components is exquisitely sensitive to detergent conditions. Our study and theirs differ in terms of experimental details, including the use of different detergents for membrane solubilization, different locations of the Flag affinity tag, different cell lines for protein expression, and a higher concentration of protein complexes in our overexpression system compared to theirs. These variations may contribute to the seemingly conflicting findings. Our results, in which ORMDL3 is apparently associated with the complexes containing the SPTLC1 variants,

demonstrate that the unrestrained sphingolipid biosynthesis caused by SPTLC1 ALS variants is associated, at least in part, with impaired ceramide sensing of these variants when ORMDL is contained within the SPT complex. Therefore, our study indicates an important role of impaired ceramide sensing in childhood ALS development.

## Discussion

It is crucial to maintain proper sphingolipid levels for normal cellular physiology. Ceramides have been suggested to be key lipotoxic factors in human metabolic diseases, such as obesity, type 2 diabetes, and cardiometabolic disorders[32,33]. In sphingolipid-rich structures such as the epidermal permeability barrier and the myelin sheath, dysregulation of sphingolipid metabolism has catastrophic effects[20,34]. Our study gives molecular insights into how cells sense ceramide levels to regulate the sphingolipid biosynthetic pathway, highlighting the function of ceramide in sphingolipid homeostasis. These studies confirm that although the ORMDLs are often referred to as inhibitors of SPT activity, they are more accurately described as ceramide-sensitive regulatory subunits of the SPT complex.

Based on our structural observations and biochemical analyses, we proposed a working model for the homeostatic regulation of SPT-ORMDL by ceramide (Fig. 7). In this model, the SPT-ORMDL complex acts as a membrane-embedded ceramide sensor that can sense the ceramide levels in the ER. At low ceramide levels, the apo SPT-ORMDL complex remains in an active state since the N-terminus of ORMDL is highly flexible and cannot block substrate entry efficiently (Fig. 7, lower left). As the ceramide level increases, the binding of ceramide to

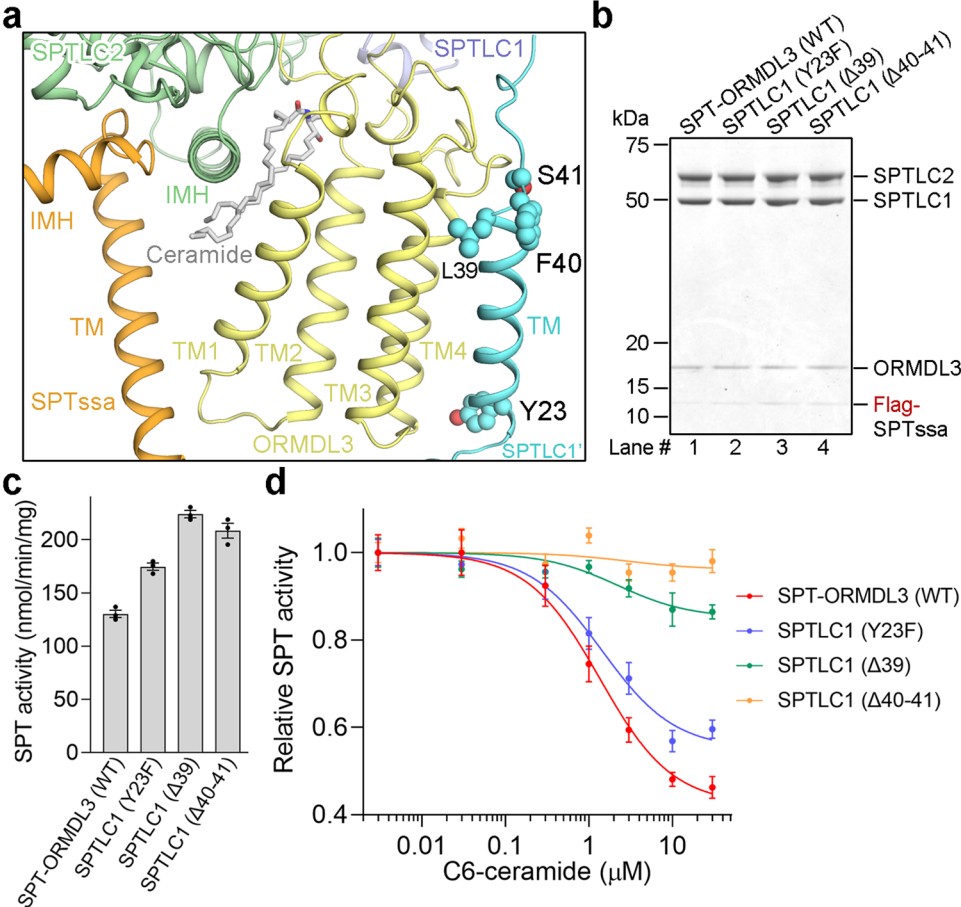

**Fig. 6 | The impaired ceramide sensing by ALS-associated SPTLC1 variants. a** A close-up view of the ALS-associated SPTLC1 variants. **b** The SPTLC1 ALS variants exhibit little effect on ORMDL3 binding. The SPT-ORMDL3 complexes bearing ALS variants, purified by size exclusion chromatography, were visualized by Coomassie-blue stained SDS-PAGE gel. **c** Functional characterization of the ALS variants by SPT activity assay. Data are presented as mean values ± SEM of three independent experiments. **d** The impaired C6-ceramide-mediated SPT-ORMDL3 activity inhibition for the ALS variants. Data are presented as mean values ± SEM of three independent experiments. Source data are provided as a Source Data file.

SPT-ORMDL can induce and lock the N-terminus of ORMDL into an inhibitory conformation, thus preventing substrate entry and turning the complex into an inactive state (Fig. 7, lower right). Although we believe this is the major mode of ORMDL-mediated inhibition of SPT, we observe that in the native membrane environment mutants containing deletion of the ORMDL3 N-terminus retain a small, but measurable, inhibition in response to ceramide. This may indicate that ceramide-bound ORMDL modulates other aspects of the SPT structure to affect SPT activity, for example by the close association with the small SPT subunit or the transmembrane helix of SPTLC1. Another possible explanation is that there are numerous ceramide species with diverse chain lengths and saturation levels in native membranes, while only a short chain C6-ceramide was employed for the in vitro assays. The interactions between SPT-ORMDL3 complex and various ceramide species might differ slightly depending on chain lengths and saturation levels, thus causing a different effect on the mutants in cells versus in vitro. Additionally, the TLC analysis, in which ceramide-binding mutants only partially reduce the ceramide content of the purified complex, may indicate the existence of additional ceramide binding sites. These could potentially mediate ceramide-induced suppression of the mutants in native membranes while bypassing the current ceramide binding site. Further studies are needed to elucidate the complexities of ceramide regulation of the SPT-ORMDL complex in vivo.

The ALS-associated SPTLC1 variants cannot sense ceramide levels properly, thus leading to unregulated SPT activity and unrestrained sphingolipid biosynthesis (Fig. 7, upper right). The ALS-associated SPTLC1 variants, located near the transmembrane interface between SPTLC1 and ORMDL3, might slightly modulate the ORMDL conformation in the SPT-ORMDL complex that allosterically alters the ceramide pocket and/or the conformation of the ORMDL N-terminus, resulting in attenuated ceramide binding and/or ceramide-mediated SPT-ORMDL inhibition. The molecular details of how the ALS-associated SPTLC1 variants lead to impaired ceramide sensing require further investigation.

The Asn13 of ORMDL3, a dominant residue responsible for the coordination of the polar head of ceramide (Fig. 3b), is strictly conserved throughout the eukaryotic ORMDL/ORM proteins (Supplementary Fig. 10a). The ceramide sensing mechanism proposed here might represent a paradigm that exists in other SPT-ORMDL/ORM systems as well. In accordance with this, mutation of Asn13 of ORMDL1 or ORMDL2 to alanine partially relieved the SPT inhibition by ORMDL and nearly eliminated the C6-ceramide-mediated SPT-ORMDL inhibition in the purified protein (Supplementary Fig. 10b–d). Notably, in addition to the well-established control of yeast ORMs by phosphorylation, the yeast SPT/ORM complex is directly responsive to ceramide in membrane preparations, similar to the mammalian ORMDLs[5]. Further studies will be required to examine whether the ceramide sensing mechanism is thoroughly conserved across phyla.

Due to the insoluble nature of native long-chain ceramides[35], we have used the short-chain ceramide analogs to examine the responses of SPT-ORMDL complexes to ceramide. Our observation that ORMDL3 being more responsive to C6-ceramide than ORMDL1 and ORMDL2

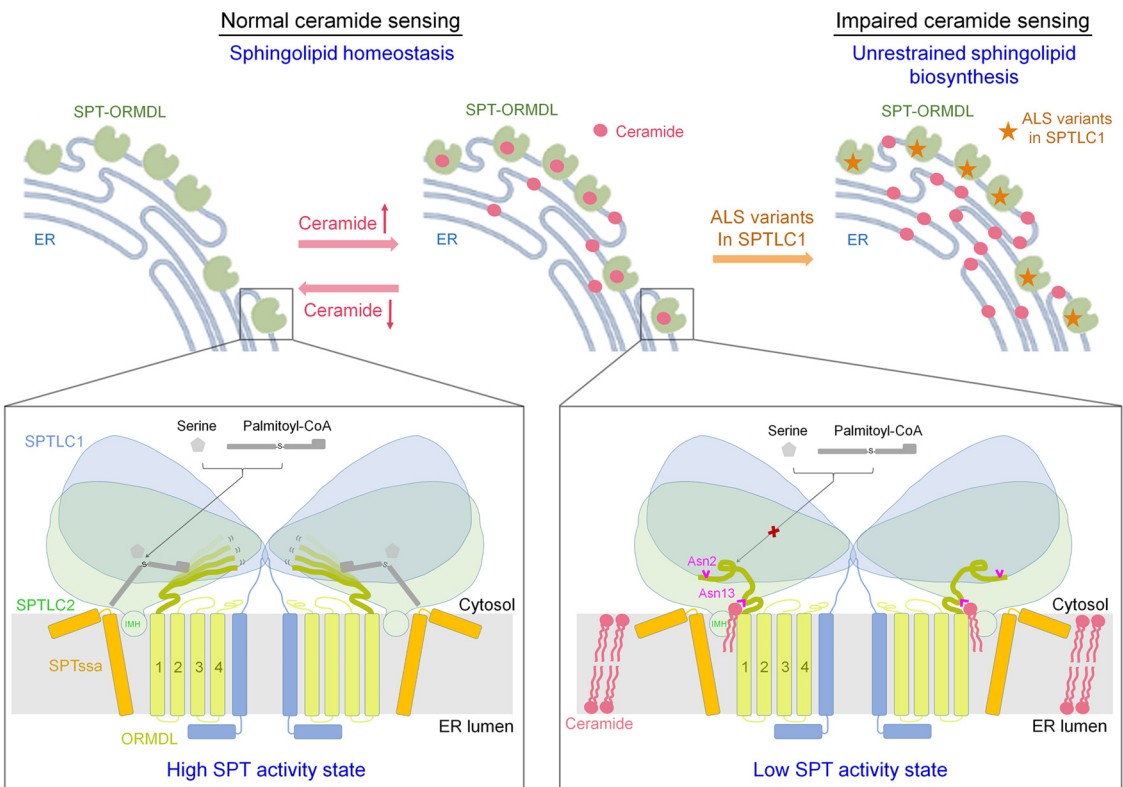

**Fig. 7 | A working model for the homeostatic regulation of SPT-ORMDL complex by ceramide and biological implications of the ALS-associated SPTLC1 variants.** At low ceramide levels, the ORMDL N-terminus remains flexible, thus allowing the entry of substrate for SPT catalysis. The constant reaction of SPT-ORMDL would lead to the accumulation of ceramide, which would inhibit SPT-ORMDL by locking the N-terminus of ORMDL3 into an inhibition conformation. The fixed ORMDL N-terminus would block the entry of substrate for further catalysis.

The Asn2 in ORMDL is important for maintaining the inactive conformation of the SPT-ORMDL complex. The Asn13 in ORMDL is critical for ceramide binding. The ALS-associated SPTLC1 variants, located near the transmembrane interface between SPTLC1 and ORMDL, impair the ceramide sensing of the SPT-ORMDL mutants. The consequently unregulated SPT activity of ALS variants would result in unrestrained sphingolipid biosynthesis.

requires confirmation in future studies that these differences are retained with natural long-chain ceramides. One possibility is that cells with different sphingolipid metabolic requirements utilize a mix of ORMDLs that establishes an optimal ceramide control of sphingolipid metabolism. Additionally, considering the natural ceramides containing distinct fatty acyl chain lengths (usually ranging from C14 to C26) exert fundamentally divergent effects[36,37], it would be fascinating to examine whether the three ORMDL isoforms can preferably respond to different ceramides based on the fatty acyl chain lengths or other structural differences.

A recent publication reported that ORMDL levels are regulated by S1P/S1P receptor-mediated control of ORMDL degradation in vascular endothelial cells[26]. This observation, in concert with our demonstration of a direct regulation by ceramide of SPT activity, indicates that SPT activity is homeostatically controlled at multiple levels, both direct and post-translational. This is reminiscent of the tight, multilevel control of sterol biosynthesis. In each of these lipid biosynthetic pathways, the products are both essential but deleterious in excess and nature has devised multiple sensing mechanisms to maintain biosynthesis, when necessary, but to limit excess production to maintain cellular integrity.

## Methods
### Protein expression and purification
The SPT and SPT-ORMDL complexes were recombinantly over-expressed and purified as previously reported[26]. Briefly, the codon-optimized full-length cDNAs for human SPTLC1, SPTLC2, SPTssa, and ORMDLs were individually subcloned into the pCAG expression vector with an N-terminal Flag tag or without a tag. A list of primers used in

this study has been supplied in Supplementary Table 3. Human embryonic kidney (HEK) 293 F suspension cells (Invitrogen) were cultured in SMM 293T-II medium (Sino Biological Inc.) and transiently transfected with the plasmid mixtures by polyethyleneimine (PEI) (YEASEN) at a cell density of $2.0–2.5 × 10^6$ cells per ml at 37 °C. After transfection for 12 hours, 10 mM sodium butyrate was applied to increase protein expression. The cells were further cultured for 48 h before being collected and resuspended in buffer containing 25 mM HEPES (pH 7.5), 150 mM NaCl, and protease inhibitor cocktail (Amresco). The membrane was solubilized with 1% (w/v) GDN (Anatrace) at 4 °C for 2 h. After centrifugation at $20,000 × g$ for 1 h, the supernatant was collected and applied to anti-Flag G1 affinity resin (GenScript). The resin was rinsed with buffer A containing 25 mM HEPES (pH 7.5), 150 mM NaCl, and 0.01% GDN and eluted with buffer A plus 200 μg/ml Flag peptide. The concentrated eluent was further purified by size-exclusion chromatography (SEC, Superose® 6 10/300 GL, GE Healthcare) with buffer A. The peak fractions were collected and concentrated for biochemical or cryo-EM studies.

### SPT activity assay
SPT activity was measured similarly as previously described using a continuous spectrophotometric assay by monitoring the release of CoA-SH, which reacts with 5,5′-dithiobis-2-nitrobenzoic acid (DTNB, $ε_{412} = 14,150\ M^{-1}·cm^{-1}$)[27,38]. The assays were performed in 96-well plates on a 100-μl or 50-μl scale at 37 °C using a BioTek plate reader. Measurements were taken at 412 nm over one hour. A typical experimental sample contained 0.1 μM SPT or SPT-ORMDL, 25 μM PLP, 0.4 mM DTNB, 10 mM L-serine, and 100 μM palmitoyl-CoA in buffer A. To measure the curve of SPT activity versus L-serine concentration, the

palmitoyl-CoA concentration was fixed at 100 μM. To measure the curve of SPT activity versus palmitoyl-CoA concentration, the L-serine concentration was fixed at 10 mM. The SPT activity was calculated from the data that falls in the linear range. Statistical analysis was performed using GraphPad Prism 8. For all curves and dot-plot graphs, each data point is the average of three independent experiments, and error bars represent the SEM. The SPT activity versus L-serine concentration was fitted with the Michaelis-Menten equation. The SPT activity versus palmitoyl-CoA concentration was fitted with an allosteric sigmoidal equation.

To measure the SPT activity inhibition curves of SPT or SPT-ORMDL complexes by C6-ceramide (Avanti, 860506 P), each protein was incubated with various concentrations of C6-ceramide on ice for 1 hr before reaction initiation. C6-ceramide was dissolved in ethanol at concentrations ranging from 0.3 μM to 3 mM as 100 × stocks. The SPT activity was normalized relative to that of the SPT or SPT-ORMDL complexes with a very low concentration of C6-ceramide. The SPT activity inhibition curves were fitted using nonlinear regression with the following equation: $Y = Bottom + (1-Bottom)/(1 + (X/IC_{50}))$. All the $IC_{50}$ values were summarized in Supplementary Table 1. The activity of SPT or SPT-ORMDL complexes in the presence of different sphingolipid species or analogs was measured similarly with a final compound concentration of 10 μM. The sphingolipid species or analogs were dissolved in ethanol at 1 mM as 100 × stocks. The SPT inhibitor myriocin was dissolved in DMSO at 1 mM as a 100 × stock. C6-dihydroceramide and C6-1-deoxyceramide were purchased from Cayman; 3-keto-dihydrosphingosine and myriocin were purchased from Glpbio; dihydrosphingosine was purchased from Sigma-Aldrich; sphingosine and sphingosine-1-phosphate were purchased from Avanti.

## Mass spectrometry and thin-layer chromatography (TLC) analysis

For LC-MS/MS, chloroform and methanol were mixed with 1 ml of SPT-ORMDL3 (0.4 mg in total) with a volume ratio of 0.5:1:1 for lipid extraction. The mixture was sonicated in an ice water bath for 30 min before centrifugation at 188 g at 4 °C for 10 min. The lower chloroform phase was collected and dried under a nitrogen stream. The lipids were dissolved in 50 μl of acetonitrile (ACN)/isopropyl alcohol (IPA)/$H_2O$ (65:30:5, v/v/v) solution containing 5 mM ammonium acetate. After sonication for 30 min and centrifugation at 13,000 × $g$ for 10 min, the lipid solution was ready for mass spectrometry analysis. About 5 μl sample was injected into the LC-MS/MS system (Q Exactive Orbitrap Mass Spectrometer, Thermo Scientific) equipped with a C18 reversed-phase column (1.9 μm, 2.1 mm × 100 mm, Thermo Scientific). Mobile phases A and B were ACN/$H_2O$ (60:40, v/v), and IPA/ACN (90:10, v/v), respectively, both supplemented with 10 mM ammonium acetate. The elution gradient started with 32% B for 1.5 min and was linearly increased to 85% B at 15.5 min. Then the gradient was held at 97% B between 15.6-18 min before being decreased to 32% B at 18.1 min for column equilibration until 22 min. The flow rate was 0.26 ml/min. The mass spectrometer was operated in negative mode with a spray voltage of 3.5 kV. Full scan mass spectrometry data were recorded from the m/z range of 166.7 – 2000. The MS/MS spectra were searched against the lipid database by LipidSearch software (Thermo Scientific). The peaks for ceramide ions, with retention time between 10.0–15.0 min, from the m/z range of 550–710 were exhibited.

For TLC, the lipids were extracted similarly as above from 0.3 mg purified protein complexes and resuspended in 10 μl of chloroform and all the sample was applied onto a TLC silica gel 60 $F_{254}$ plate (Merck). The plate was developed with a solvent system containing chloroform and methanol (190:10, v/v) by two consecutive runs in the same direction. The TLC plate was dried and stained with iodine vapor for overnight to visualize the lipids.

## Cryo-EM sample preparation and data collection

Quantifoil Cu R1.2/1.3 300 mesh grids were glow-discharged at 15 mA for 30 s in a PELCO easiGlow device before sample preparation. For the SPT-ORMDL3 (apo WT, ORMDL3-ΔN2, and ORMDL3-N13A) cryo-EM samples, 3 μl aliquots of purified protein at ~18 mg/ml were applied to the glow-discharged grids. For the C6-ceramide-bound SPT-ORMDL3 sample, ~18 mg/ml WT SPT-ORMDL3 complex was preincubated with 1 mM C6-ceramide (10 mM stock dissolved in ethanol) on ice for 1 hr before being applied to the grids. The grids were then blotted for 4.5 s and flash-frozen in liquid ethane cooled by liquid nitrogen using Vitrobot (Mark IV, Thermo Fisher Scientific).

All the datasets were automatically collected with SerialEM[39] on a Titan Krios microscope operated at 300 kV with a K2 Summit direct electron detector (Gatan) and a GIF Quantum energy filter (Gatan) with a slit width of 20 eV. The movie stacks were collected at a nominal magnification of ×130,000 with defocus values from −2.0 to −1.0 μm. Each stack was exposed in super-resolution mode for 5.76 s in 32 frames. The total dose for each stack was 50 e⁻/Å². The stacks were motion-corrected using MotionCor2[40] with a binning factor of 2, resulting in a pixel size of 1.08 Å, meanwhile, dose weighting was performed[41]. The defocus values were estimated by Gctf[42].

## Cryo-EM data processing

For the C6-ceramide-bound SPT-ORMDL3 complex dataset, a total of 886,714 particles were automatically picked from 1944 micrographs in Relion 3.0[43]. The 2D classification resulted in 755,364 good particles that were subjected to a global search 3D classification. A total of 511,642 particles were selected and subjected to 3D auto-refinement. The refined particles were then subjected to local search 3D classifications with or without a monomer mask, respectively. The local search 3D classification without mask generated two good 3D classes with relative movements between the protomers, accounting for 187,017 and 149,014 particles, respectively. 3D auto-refinement of the particles with overall mask and C2 symmetry yielded reconstructions with overall resolutions of 3.3 Å and 3.2 Å, respectively. The local search 3D classification with monomer mask separated 337,730 particles with the best features. 3D auto-refinement of the particles with a monomer mask resulted in a reconstruction with an overall resolution of 2.9 Å.

The procedures for SPT-ORMDL3 (apo WT, ORMDL3-ΔN2, and ORMDL3-N13A) data processing were similar as above. Briefly, 928,652/1,073,957/839,572 particles were automatically picked from 1,798/1,835/1,410 micrographs of apo SPT-ORMDL3, SPT-ORMDL3 (ORMDL3-ΔN2), and SPT-ORMDL3 (ORMDL3-N13A), respectively. 2D classifications resulted in 789,770/951,219/542,395 good particles that were then subjected to global search 3D classifications and auto-refinements. Further local search 3D classifications without masks also generated two good 3D classes with relative movements between the protomers for each SPT-ORMDL3 complex at resolutions of 3.0–3.4 Å. Local search 3D classifications with a monomer mask yielded 358,765/317,358/249,589 particles, respectively. 3D auto-refinements of the particles with monomer mask resulted in reconstructions with overall resolutions of 2.7 Å, 3.1 Å, and 2.9 Å for apo SPT-ORMDL3, SPT-ORMDL3 (ORMDL3-ΔN2), and SPT-ORMDL3 (ORMDL3-N13A), respectively.

All the 2D classifications, 3D classifications, and 3D auto-refinements were performed in Relion 3.0. Resolutions were estimated by the gold-standard Fourier shell correlation 0.143 criterion[44] with high-resolution noise substitution[45].

## Model building and refinement

The previously reported human SPT-ORMDL3* complex structure (PDB 6M4O) was used as the initial model for model building. The model was docked into each map using Chimera[46] and every residue was manually adjusted with Coot[47]. Structure refinements were done

using Phenix[48] in real space with secondary structure and geometry restraints. All the structural models were validated using Phenix and MolProbity[49]. The refinement and validation statistics were summarized in Supplementary Table 2. All structural figures were prepared using PyMol[50] or UCSF Chimera.

## Membrane-based assay

**Cell culture.** HeLa ORMDL2/3 knockout cells were cultured in Dulbecco's minimal essential medium (Gibco, Cat# 11960-044) with 10% fetal bovine serum (Gemini Biochem Cat# 900-108) supplemented with 1% L-glutamine (Cat# 25030-061) and 1% Penicillin/Streptomycin (Cat# 215140-122).

**Generation of ORMDL2/3 knockout cell line.** The ORMDL2/3 knockout out cell line was generated by the CRISPR/Cas9 technique. In brief, guide RNAs were designed against the exon two and exon three of ORMDL2 and ORMDL3 isoform respectively. Guide RNAs were subcloned into the lentiviral plasmid CrisprV2 (Addgene #52961). Lentiviral particles were produced by transfection of HEK293T cells expressing the SV40-T antigen with PEI along with the packing plasmids psPAX2 (Addgene #12260) and pMD2.G (Addgene #12260). Supernatants containing lentiviral particles were collected, filtered, and brought to 10% PEG-8000 (Sigma #89510), incubated on ice overnight, and pelleted. The viral pellets were resuspended in Optimem (Invitrogen) and stored at −80 °C. HeLa cells were then infected with the virus particles by pelleting in the presence of polybrene (8 µg/ml). In total, 48 h later cells were selected by puromycin treatment. The single clones were isolated and amplified for the screening of positive clones. Sanger sequencing was used to verify the successful knockout of ORMDL2/3. Two ORMDL2/3 knockout clones were validated for the presence of only ORMDL1 by immunoblotting that siRNA directed against ORMDL1 completely removed ORMDL immunoreactivity.

**siRNA and plasmid transfection.** HeLa cells were plated at $4 \times 10^6$ per p150 plate in Penicillin/Streptomycin free cell culture media. The next day cells were transfected with 10 µMol/plate siRNA against ORMDL1 [Integrated DNA Technologies (ID# hs.ri.ORMDL1.13.4)] using Lipofectamine RNAi Max (ThermoFisher Cat# 13778150) according to the manufacturer's protocol. Penicillin Streptomycin free media was used to culture the transfected cells. The next day, cells were either transfected with 11 µg of single chain SPT (scSPT)[28] and 11 µg pCMV XL5 vector (as a vector control for human ORMDL3) or 11 µg of scSPT and 11 µg of human ORMDL3 WT or mutant constructs using PEI transfection reagent per p150 plate. Cells were harvested 48 hours after plasmid transfection for total mammalian cell membrane isolation as described later.

**Preparation of HeLa total membranes.** After siRNA and plasmid transfection, cells were harvested by trypsinization and washed with 1× cold PBS. Cell pellets from one p150 plate were resuspended in a 3.2 ml ice-cold swelling buffer (10 mM Tris pH 7.5, 15 mM KCl, 1 mM MgCl₂) and incubated on ice for 15 mins. Next, 1 ml 1 M sucrose, 160 µl 25× protease inhibitor, 14 µl 200 mM EDTA pH 7.5 was added to the resuspended cells. The cells were broken by using a ball bearing homogenizer (10 strokes). Unbroken cells were removed by centrifugation at 72 g for 10 mins at 4 °C. Supernatants were collected and centrifuged at $417{,}000 \times g$ for 25 min at 4 °C in a TLA 110 rotor (Beckman). Pellets were resuspended in 350 µl of resuspension buffer (25 mM Tris pH 7.5, 250 mM sucrose, 1× protease inhibitor) and passed through 16-gauge needle, then 19-gauge needle, then 22-needle and finally through 26-gauge needle. Isolated membranes were aliquoted as 50 µl aliquots, snap frozen in liquid nitrogen and stored at −80 °C.

**SDS-PAGE and immunoblotting.** Protein concentration was measured in the isolated membranes and a 5× Laemmli sample buffer was added to the samples. Samples were incubated at 60 °C for 1 h and were electrophoresed on Tris-SDS gradient gels. Proteins were transferred to the activated PVDF membranes and blots were incubated with blocking buffers, either with 5% non-fat milk or 5% fat-free BSA in 1× TBST for 2 h. The blots were incubated with anti-SPTLC1 (Beckton Dickinson Cat#611305) (1:3000 in 5% non-fat milk), anti-Sec22b (Santa Cruz Biologicals Cat #Sc-101267) (1:250 in 5% non-fat milk), anti-Calnexin (Enzo Life Sciences Cat #ADI-SPA-860-F) (1:3000 in 5% non-fat milk) and anti-ORMDL (Sigma-Millipore Cat #ABN417) (1:3000 in 5% fat-free BSA) for overnight with continuous shaking at cold. Next day membranes were washed 3× for10 mins with 1× TBST at room temperature. HRP conjugated secondary antibodies, anti-mouse and anti-rabbit, were diluted in blocking buffers at 1:10,000 dilutions, and were added to blots for overnight incubation with continuous shaking at cold temperatures. Next day, blots were washed 3× for 10 min with 1× TBST and with Milli Q water for 15 mins. Blots were then visualized with ECL-Plus (Thermo-Fisher) according to the manufacturer's instructions using HyblotCL film (Thomas Scientific).

**In vitro SPT activity measurement in the isolated membranes.** Assays were performed in 2 ml microcentrifuge tubes essentially as described[7]. 5 µg membranes were preincubated, in a total volume of 100 µl consisting of 50 mM HEPES, 1 mM DTT, 2 mM EDTA, 40 µM pyridoxal 5′-phosphate for 40 mins on ice. A 1 mM C8-ceramide stock solution was prepared by adding 20 mM C8-ceramide (in methanol) to 5% fatty acid-free BSA in 1 × PBS. Dilutions were made in 5% fatty-acid free BSA in 1 × PBS to achieve the desired concentrations. The assay was initiated by addition of 100 µl of a prewarmed pre-mix containing 2 mM L-serine, 100 µM palmitoyl-CoA and 10 µCi of [³H-Serine] either in the presence of control (5% fatty-acid free BSA in 1 × PBS), C8-ceramide at indicated concentration or 1 µM myriocin for 1 hr at 37 °C. The assay was terminated by the addition of 400 µl alkaline methanol (0.7 gm KOH/100 ml methanol) and vortexing. 100 µl of chloroform was added to the tubes followed by vortexing and brief centrifugation. 500 µl of chloroform, 300 µl of alkaline water (100 µl 2 N NH₃OH + 100 ml water) and 100 µl of 2 N NH₃OH to break the phases. Samples were vigorously vortexed and centrifuged at 17,000 g for 1 min. The upper, aqueous layer was aspirated off. The lower, organic phase was washed twice with 1 ml alkaline H₂O by vigorous vortexing, centrifugation and aspirating off the upper phase. 300 µl of the organic phase was collected and was dried in the scintillation vials under N₂ gas. 5 ml Betamax was added to the dried samples, vortexed and vials were counted for 5 min in Beckman-Coulter LS6500. C8-ceramide (Cat # 860508) and myriocin (Cat # 63150) were purchased from Avanti polar and Cayman Chemicals, respectively. L-[³H]-serine was purchased from American Radiochemical Corporation. Palmitoyl-CoA (Cat # P9716-10MG), DTT (Cat # D0632), pyridoxal 5′-phosphate (Cat # P9255) and L-serine (Cat # S4500) were purchased from Sigma. Chloroform (Cat # C297-4), methanol (Cat #A452-4), EDTA (Cat # S311-500), potassium hydroxide (Cat # P250-500) and protease inhibitor (Cat # A32963) were purchase from Thermo Fisher. Ecolite-Scintillation fluid (Cat # 882475) and Beta max-scintillation fluid (Cat # 880020) was purchased from MP biomedicals.

**Statistics and reproducibility.** The sample size in this study was not predetermined using statistical methods, and the experiments were not randomized. The investigators were also not blinded to allocation during both experiments and outcome assessment. All the experiments were conducted independently at least three times, yielding comparable results.

## Reporting summary

Further information on research design is available in the Nature Portfolio Reporting Summary linked to this article.

## Data availability

The EM density maps generated in this study have been deposited in the EMDB under accession codes EMD-33864 (SPT-ORMDL3 + C6-ceramide), EMD-33866 (apo SPT-ORMDL3), EMD-33868 (ORMDL3-ΔN2 variant), and EMD-33869 (ORMDL3-N13A variant). The atomic coordinates have been deposited in the PDB under the accession codes 7YIU (SPT-ORMDL3 + C6-ceramide), 7YIY (apo SPT-ORMDL3), 7YJ1 (ORMDL3-ΔN2 variant), and 7YJ2 (ORMDL3-N13A variant). Source data are provided with this paper.

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

## Acknowledgements

We thank the Cryo-EM Facility of the Southern University of Science and Technology (SUSTech) for providing the facility support. We are grateful to Ma, X., Gao, Y., and all the other staff members in the SUSTech Cryo-EM center for their technical support on cryo-EM data collection. The mass spectrometry data for endogenously bound ceramides were obtained using equipment maintained by SUSTech Core Research Facilities. This work was supported by the National Natural Science Foundation of China (32122043 and 92057101 to X.G.), the Guangdong Basic and Applied Basic Research Foundation (2019B151502047 to X.G.), the Shenzhen Science and Technology Program (RCYX20200714114522081 and 20220815111002002 to X.G.), and the National Institutes of Health (R21NS120128 to B.W.W.).

## Author contributions

X.G. conceived and supervised the project. X.G., T.X., and P.L. designed experiments. T.X. and P.L. performed the in vitro biochemical and structural experiments with the help of X.W., F.D., Z.Z., J.Y., Q.F., and W.L. B.W.W., U.M., F.F., and H.V. contributed to the membrane-based assay. All authors contributed to the data analysis. T.X. contributed to manuscript preparation. X.G. wrote the manuscript. T.X. and B.W.W. revised the manuscript.

## Competing interests

The authors declare no competing interests.
