## [Peer Review File · Nature Communications]

Ceramide sensing by human SPT-ORMDL complex for establishing sphingolipid homeostasisREVIEWER COMMENTS

Reviewer #1 (Remarks to the Author):

This is an excellent study which defines the mechanism by which SPT-ORMDL senses ceramide. The study is comprehensive and performed at an outstanding level of technical competence. Having said that, some of the current study was based on a mistake in a previous experiment where a cloning issue led to the use of a wrong plasmid. In my opinion, this previous mistake does not take away from the importance of the current study and indeed the authors are to be commended for clearing up the matter of the ORMDL* plasmid at the beginning of their study.

One issue which I find quite annoying is the huge amount of extended data, much of which appears essential for the flow of the paper. If I understand correctly, Nature Communications allows up to 10 display items, and I would like to see some of the extended data moved to the main text, such as Fig. 1, 3, 8 and 10. I believe this would help the reader follow the story much better.

Other points include:

1. The use of C6-ceramide is justified because of its higher solubility. However, have the authors tried to model in different chain length ceramides using molecular modeling? This may be what extended Fig. 6a is trying to show but I am not sure. More discussion of the effect of different acyl chains, and how they bind, would be helpful.
2. I find Fig. 1a a bit confusing inasmuch as the ORMDLs are part of the SPT complex but in the figure is looks like they are added to the complex like ceramides.
3. Fig. 5 is an excellent summary but I feel it can be more visually attractive, i.e the artwork can be better

Reviewer #2 (Remarks to the Author):

This groundbreaking study provides important mechanistic insight into how SPT-ORMDL, the rate limiting enzyme in sphingolipid biosynthesis, senses ceramides to establish sphingolipid homeostasis. The authors previously reported cryo-EM structures of the enzyme complex and used semi-reconstituted systems (permeabilized cells, isolated membranes) to support a role of ceramide as the downstream product that mediates feedback inhibition. However, the structural basis for SPT-ORMDL regulation by ceramide remained to be established. In the present study, the authors acquired novel cryo-EM structures of SPT-ORMDL in which they were able to visualize a lipid-like density resembling a ceramide molecule. Using structure-guided mutational analysis of the putative ceramide binding site followed by detailed structural and functional studies, the authors uncovered a crucial role of the highly flexible N-terminus of ORMDL in ceramide-mediated feedback regulation. Their findings indicate that ceramide binding induces and locks the N-terminal region into an inhibitory conformation that would prevent substrate entry into the enzyme complex. The study also includes data suggesting that unrestrained sphingolipid biosynthesis caused by SPTLC1 variants implicated in childhood amyotrophic lateral sclerosis (ALS) is associated, at least in part, with impaired ceramide sensing.

Overall, I find the work both compelling and of broad interest. The model proposed by the authors is plausible and well supported by high quality data. However, as outlined below, I believe the functional characterization of putative ceramide binding mutants is incomplete. In addition, the authors need to cite and discuss a recent study in which the unrestrained sphingolipid biosynthesis mediated by SPTLC1 ALS variants is ascribed to an impaired ORMDL protein binding; this concept is distinct from the one proposed by the authors.

1) The structure of a ceramide-bound SPT-ORMDL3 complex enabled the authors to map residues potentially involved in ceramide binding. They propose that Asn13 of ORMDL3, which is conserved among ORMDL/ORM proteins across the eukaryotic kingdom, coordinates the polar head of ceramide (Fig. 2c and Fig. 5). Indeed, mutation of Asn13 into alanine caused a partial disappearance of the ceramide-like density in the structure of the corresponding SPT-ORMDL3 variant (Fig. 4a) and virtually abolished the inhibitory effect of C6-ceramide on its *in vitro* SPT activity (Fig. 2e). However, direct evidence that Asn13 is critical for ceramide binding is lacking. To overcome this, the authors should subject the purified SPT-ORMDL3-N13A variant to lipid extraction and TLC analysis as was done for the wild type complex (Extended Data Fig. 6d). Alternatively, they could compare the ability of wild type and mutant complexes to bind externally added C6-ceramide. Using a commercial clickable short-chain ceramide in combination with quantitative fluorescence TLC analysis should enable them to obtain quantitative data on ceramide binding for both SPT-ORMDL3-N13A and SPT-ORMDL3-deltaN2, hence verifying their structural data that indicate a disrupted ceramide binding (Figs. 3e and 4a).

2) A previous study showed that the SPTLC1 variants delta39 and delta40-41 that cause childhood amyotrophic lateral sclerosis (ALS) result in unrestrained sphingolipid biosynthesis (Mohassel et al. 2021;

Ref. 30). In the present study, the authors find that these mutations have little impact on ORMDL3 binding yet attenuate the inhibitory effect of C6-ceramide on in vitro SPT activity (Fig. 4d-f). Based on these results, they postulate an important role of impaired ceramide sensing in the development of childhood ALS. However, Lone et al. recently reported that the dysfunctional feedback regulation and increased SPT activity of SPTLC1-ALS variants is primarily due to an impaired ORMDL binding (Lone et al. 2022, JCI 132, e161908; PMID: 35900868). It would be appropriate that the authors cite the study by Lone et al. and discuss potential grounds for these seemingly conflicting findings.

3) p. 5: the authors state that the ORMDL3 plasmid used in their previous study (named ORMDL3*; Li et al., 2021; Ref. 27) contains an extra start codon and 11 other nucleotides upstream of the protein's ORF. Expression of ORMDL3* yields a minor fraction of full-length ORMDL3 and major fraction of a truncated protein that lacks the N-terminal 16 residues (ORMDL3-deltaN17). In the present study they used an alternative ORMDL3-FL from which these extra nucleotides were removed and that yields only the full-length protein. To avoid any confusion, it would be very helpful if the authors include the relevant nucleotide and corresponding protein sequences of both ORMDL3* and ORMDL3-FL in panel A of Extended Data Fig. 1.

Reviewer #3 (Remarks to the Author):

Ceramide lipids play a variety of vital roles in cellular and organism homeostasis, and its dysregulated synthesis can lead to catastrophic consequences. Activity of the serine palmitoyltransferase (SPT) enzyme, which is responsible for initiating the rate-limiting step in sphingolipid biosynthesis, is modulated by its association with ORM/ORMDL proteins and in a feedback loop by cellular levels of ceramide lipids. However, the structural basis for this feedback loop remains poorly understood. Here, the authors use a combination of functional assays, cryoEM structural experiments and mutagenesis to investigate this question. They show that SPT-ORMDL complex co-purify with endogenous ceramide lipids, likely a long chain C24 ceramide, and mutagenesis show that this ceramide binding site plays a key role in suppressing SPT activity. In agreement with past results, the N-terminus of ORMDL3 interacts with SPTLC1 and SPTLC2 and likely interferes with their catalytic activity. Remarkably, the authors show that mutations in the N-terminus destabilize ceramide binding to the inhibitory site and conversely, mutations at the inhibitory binding site destabilize the N-terminus, thus establishing an allosteric mechanism between these two inhibitory regions. Importantly, the authors show that SPTLC1 mutations associated with childhood ALS cause impaired ceramide sensing in the SPT-ORMDL3 complex, providing a mechanistic link between this disease and sphingolipid biosynthesis.

Overall, this is an excellent manuscript. The experiments are well-designed, logically laid out, and appropriately interpreted. This was one of the rare cases when, while reading a manuscript and thinking

of the next experiment, I'd satisfactorily find it in the following section. As such, I have no major concerns, only a couple of minor suggestions and a point to perhaps consider in the discussion.

I realize it is a big ask, but the manuscript would be greatly strengthened if the authors could determine the cryoEM structure of one of the ALS mutants. This would provide a structural framework to interpret their interesting functional results.

The finding that the effects of the mutants is somewhat blunted when assayed in cells compared to when measured in vitro is interesting. I wonder if this could reflect the fact that in native membranes multiple different kinds of ceramides will be delivered to the complex, whereas in the in vitro experiments only a single type is considered. If the interactions between the SPT-ORMDL3 complex and various ceramide lipids differ slightly depending on chain length and saturation, then it is possible that the mutants might have a different effect in vitro or in cells. If appropriate, this could be added to the discussion.

I would suggest the authors explain in a little more detail the rationale for the mutagenesis in Fig. 2. Are the residues considered the only ones with side chains within interaction distance? Was a cut-off distance used in their analysis? if so, this should be stated. For example, in Fig. 2c it appears that I22 (from ORM DL3) is quite far from the ceramide tail whereas F498 (from SPTLC2) seems close. However, the authors only mutated the former and not the latter.

The structural shifts in Fig. 4b are quite difficult to make out. I would suggest that the authors use colors with a bit more contrast than light gray and pale yellow for the two structures, and/or color the ceramide molecule in a color different from the light gray used for the WT SPT-ORMDL3.

Response to Reviewers:

Reviewer #1:

This is an excellent study which defines the mechanism by which SPT-ORMDL senses ceramide. The study is comprehensive and performed at an outstanding level of technical competence. Having said that, some of the current study was based on a mistake in a previous experiment where a cloning issue led to the use of a wrong plasmid. In my opinion, this previous mistake does not take away from the importance of the current study and indeed the authors are to be commended for clearing up the matter of the ORMDL plasmid at the beginning of their study.*

One issue which I find quite annoying is the huge amount of extended data, much of which appears essential for the flow of the paper. If I understand correctly, Nature Communications allows up to 10 display items, and I would like to see some of the extended data moved to the main text, such as Fig. 1, 3, 8 and 10. I believe this would help the reader follow the story much better.

We thank the reviewer for his/her appreciation of our work. As suggested by this reviewer, we've moved some of the extended data to the main text. Several panels from the original Extended Figs. 1 (panels a and b of original Extended Figure 1), 4 (panels a and b of original Extended Figure 4), and 7 (panel c of original Extended Figure 7) are now incorporated into the Figs. 1, 2, and 5 of the revised manuscript. We agree with this reviewer that this reorganization would help the readers to follow the story much better.

Other points include:

1. The use of C6-ceramide is justified because of its higher solubility. However, have the authors tried to model in different chain length ceramides using molecular modeling? This may be what extended Fig. 6a is trying to show but I am not sure. More discussion of the effect of different acyl chains, and how they bind, would be helpful.

We appreciate this reviewer for the constructive comment. In the C6-ceramide-bound SPT-ORMDL3 structure, our analysis suggests that the ceramide density in the structure arises from the average of C6-ceramide and endogenous bound long-chain ceramide (such as C24-ceramide). We have modeled the C24-ceramide molecule into the EM map as presented in Fig. 2a. As suggested by this reviewer, we have added a detailed view of the hydrophobic interactions between C24-ceramide acyl-chain and SPT-ORMDL3 in Supplementary Fig. 5c of the revised manuscript based on this model. The discussion of the effect of different acyl chains and how they bind was also added in the revised manuscript (lines 180-187) as follows “Ceramide molecules with longer acyl chains could provide stronger hydrophobic interactions compared to short-chain C6-ceramide (Supplementary Fig. 5c). This is supported by previous data that demonstrated that as acyl-chain length increases ceramides becomes more potent at inhibiting SPT, as assayed in

isolated membranes⁵. This could account for the observation that C6-ceramide only partially displaces endogenous long-chain ceramides, resulting in a mixture of C6-ceramide and natural long-chain ceramides in the C6-ceramide-bound SPT-ORMDL3 structure (Fig. 2a).”

2. I find Fig. 1a a bit confusing inasmuch as the ORMDLs are part of the SPT complex but in the figure it looks like they are added to the complex like ceramides.

Point taken. Fig. 1a has been revised accordingly.

3. Fig. 5 is an excellent summary but I feel it can be more visually attractive, i.e the artwork can be better

Point taken. We have revised Fig. 7 (original Fig. 5), with the aim of making it more visually attractive. We hope that the changes we have made will improve this.

We thank the reviewer for his/her time and constructive comments.

Reviewer #2:

This groundbreaking study provides important mechanistic insight into how SPT-ORMDL, the rate limiting enzyme in sphingolipid biosynthesis, senses ceramides to establish sphingolipid homeostasis. The authors previously reported cryo-EM structures of the enzyme complex and used semi-reconstituted systems (permeabilized cells, isolated membranes) to support a role of ceramide as the downstream product that mediates feedback inhibition. However, the structural basis for SPT-ORMDL regulation by ceramide remained to be established. In the present study, the authors acquired novel cryo-EM structures of SPT-ORMDL in which they were able to visualize a lipid-like density resembling a ceramide molecule. Using structure-guided mutational analysis of the putative ceramide binding site followed by detailed structural and functional studies, the authors uncovered a crucial role of the highly flexible N-terminus of ORMDL in ceramide-mediated feedback regulation. Their findings indicate that ceramide binding induces and locks the N-terminal region into an inhibitory conformation that would prevent substrate entry into the enzyme complex. The study also includes data suggesting that unrestrained sphingolipid biosynthesis caused by SPTLC1 variants implicated in childhood amyotrophic lateral sclerosis (ALS) is associated, at least in part, with impaired ceramide sensing.

Overall, I find the work both compelling and of broad interest. The model proposed by the authors is plausible and well supported by high quality data. However, as outlined below, I believe the functional characterization of putative ceramide binding mutants is incomplete. In addition, the authors need to cite and discuss a recent study in which the unrestrained sphingolipid biosynthesis mediated by SPTLC1 ALS variants is ascribed to an impaired ORMDL protein binding; this concept is distinct from the one proposed by the authors.

We thank the reviewer for his/her positive comments concerning our manuscript. As suggested by this reviewer, we have made revisions to the manuscript with respect to the functional characterization of putative ceramide binding mutants and the discussion related to a recent study on SPTLC1 ALS variants. The details of these revisions are provided below.

*1) The structure of a ceramide-bound SPT-ORMDL3 complex enabled the authors to map residues potentially involved in ceramide binding. They propose that Asn13 of ORMDL3, which is conserved among ORMDL/ORM proteins across the eukaryotic kingdom, coordinates the polar head of ceramide (Fig. 2c and Fig. 5). Indeed, mutation of Asn13 into alanine caused a partial disappearance of the ceramide-like density in the structure of the corresponding SPT-ORMDL3 variant (Fig. 4a) and virtually abolished the inhibitory effect of C6-ceramide on its *in vitro* SPT activity (Fig. 2e). However, direct evidence that Asn13 is critical for ceramide binding is lacking. To overcome this, the authors should subject the purified SPT-ORMDL3-N13A variant to lipid extraction and TLC analysis as was done for the wild type complex (Extended Data Fig. 6d). Alternatively, they could compare the ability of wild type and mutant complexes to bind externally added C6-ceramide. Using a*

commercial clickable short-chain ceramide in combination with quantitative fluorescence TLC analysis should enable them to obtain quantitative data on ceramide binding for both SPT-ORMDL3-N13A and SPT-ORMDL3- Δ N2, hence verifying their structural data that indicate a disrupted ceramide binding (Figs. 3e and 4a).

We thank this reviewer for the critical comment. Per this reviewer's request, we've performed a comparative TLC analysis for ceramides co-purified with the WT SPT-ORMDL3, SPT-ORMDL3 (ORMDL3-N13A), and SPT-ORMDL3 (ORMDL3- Δ N2) complexes (as shown below, Supplementary Fig. 5e, f in the revised manuscript). The amount of co-purified ceramides in the ORMDL3-N13A and ORMDL3- Δ N2 complexes was reduced by approximately 30% and 25% compared to the WT complex. The results further support that the structurally and biochemically characterized ceramide-binding site around ORMDL3-Asn13 and the Asn2 of ORMDL3 are important for ceramide binding. This part of the analysis has been added to the main text of the revised manuscript (lines 271-273 & 286-288).

Based on the TLC analysis, the molar stoichiometry between WT SPT-ORMDL3 protomer or SPT-ORMDL3 (ORMDL3-N13A) protomer or SPT-ORMDL3 (ORMDL3- Δ N2) protomer and bound ceramide is estimated to be 1:2.94 or 1:1.94 or 1:2.22, respectively. Our data suggest that each WT SPT-ORMDL3 protomer binds with approximately three molecules of ceramide, which raises the possibility of additional ceramide binding sites other than the one observed in the current structure. Alternatively, there could be non-specific binding between the ceramide and SPT-ORMDL3 complex. Both possibilities mentioned above could explain why the ORMDL3-N13A and ORMDL3- Δ N2 mutants, despite having a ~30% and ~25% reduction in ceramide enrichment, still bind quite a bit of ceramide in the TLC analysis. These analyses have also been added to the main text of the revised manuscript (lines 194-201, 273-275, 288-289, 406-411).

The reviewer's suggestion to utilize bifunctional ceramide reagents to directly detect ceramide binding is entirely appropriate and would be extremely useful, for example, in determining binding kinetics. Indeed, we had previously attempted to use this approach but were ultimately unsuccessful. Below we illustrate an experiment in which affinity-purified SPT-ORMDL3 complex was incubated with PacFA Cer, UV activated, and clicked to a fluorescent probe. Despite obtaining excellent labeling of ORMDL3, attempts to compete it with a 10-fold excess of either C8 or C16 ceramide were unsuccessful. Without being able to competitively block labeling we do not consider this data to be a convincing demonstration of specific binding. We conducted additional experiments to try and address this issue, including the use of other ceramide probes such as pac-C7 ceramide, but none of them demonstrated specific labeling - even when using a 100-fold excess of unlabeled ceramide over the labeling reagent. Moreover, according to the study by Deng et al. (2021 JBC, PMID: 34793833), a photoactivatable pac-C7-Cer was able to bind non-specifically with SPTLC1, as excess C6-ceramide was unable to compete with pac-C7-Cer for binding. Taken together, our results and those of Deng et al. suggest that this method may be challenging for investigating the SPT-ORMDL3 complex.

Labeling of ORMDL3 in affinity-purified SPT-ORMDL3 complex with a photoactivatable/clickable ceramide analog. **a**, Chemical structure of PacFA ceramide used in the labeling experiment. **b**, Silver-stained gel of Flag resin affinity-purified SPT-ORMDL3 complex containing ORMDL3-FLAG and a single-chain construct (scSPT) of SPTLC1/SPTLC2/SPTssa. **c**, The affinity-purified SPT-ORMDL3 complex (2 μ g) was incubated with PacFA Cer (100 μ M) with or without C8 or C16-ceramide (1 mM), then photoactivated with UV light and clicked to Alexa Fluor 647.

Overall, our EM data, ceramide inhibition assays, and TLC analyses all support the notion that the ORMDL3-Asn13 is critical for ceramide binding and that the ceramide binding for both ORMDL3-N13A and ORMDL3- Δ N2 mutants was significantly disrupted.

2) A previous study showed that the SPTLC1 variants delta39 and delta40-41 that cause childhood amyotrophic lateral sclerosis (ALS) result in unrestrained sphingolipid biosynthesis (Mohassel et al. 2021; Ref. 30). In the present study, the authors find that these mutations have little impact on ORMDL3 binding yet attenuate the inhibitory effect of C6-ceramide on in vitro SPT activity (Fig. 4d-f). Based on these results, they postulate an important role of impaired ceramide sensing in the development of childhood ALS. However, Lone et al. recently reported that the dysfunctional feedback regulation and increased SPT activity of SPTLC1-ALS variants is primarily due to an impaired ORMDL binding (Lone et al. 2022, JCI 132, e161908; PMID: 35900868). It would be appropriate that the authors cite the study by Lone et al. and discuss potential grounds for these seemingly conflicting findings.

We thank this reviewer for the insightful comment. In the revised manuscript (lines 360-373), we cite the study by Lone et al and discuss the potential grounds for these seeming conflicting findings as follows: “Another study on the SPTLC1-ALS variants reported that ORMDLs had weakened interaction with the SPTLC1 (\$\Delta\$ 39) and SPTLC1 (\$\Delta\$ 40-41) variants as measured by co-immunoprecipitation and blue native PAGE, while the binding was retained with the SPTLC1 (Y23F) variant³¹. It is worth noting that the isolation of ORMDL with the other SPT complex components is exquisitely sensitive to detergent conditions. Our study and theirs differ in terms of experimental details, including the use of different detergents for membrane solubilization, different locations of the Flag affinity tag, different cell lines for protein expression, and a higher concentration of protein complexes in our overexpression system compared to theirs. These variations may contribute to the seemingly conflicting findings. Our results, in which ORMDL3 is apparently associated with the complexes containing the SPTLC1 variants, demonstrate that the unrestrained sphingolipid biosynthesis caused by SPTLC1 ALS variants is associated, at least in part, with impaired ceramide sensing of these variants when ORMDL is contained within the SPT complex. Therefore,”.

3) p. 5: the authors state that the ORMDL3 plasmid used in their previous study (named ORMDL3; Li et al., 2021; Ref. 27) contains an extra start codon and 11 other nucleotides upstream of the protein's ORF. Expression of ORMDL3* yields a minor fraction of full-length ORMDL3 and major fraction of a truncated protein that lacks the N-terminal 16 residues (ORMDL3-deltaN17). In the present study they used an alternative ORMDL3-FL from which these extra nucleotides were removed and that yields only the full-length protein. To avoid any confusion, it would be very helpful if the authors include the relevant nucleotide*

and corresponding protein sequences of both ORMDL3 and ORMDL3-FL in panel A of Extended Data Fig. 1.*

Point taken. The nucleotide and corresponding protein sequences of both ORMDL3* and ORMDL3-FL are included in Supplementary Fig. 1 of the revised manuscript.

We thank the reviewer for his/her time and constructive comments.

Reviewer #3:

Ceramide lipids play a variety of vital roles in cellular and organism homeostasis, and its dysregulated synthesis can lead to catastrophic consequences. Activity of the serine palmitoyltransferase (SPT) enzyme, which is responsible for initiating the rate-limiting step in sphingolipid biosynthesis, is modulated by its association with ORM/ORMDL proteins and in a feedback loop by cellular levels of ceramide lipids. However, the structural basis for this feedback loop remains poorly understood. Here, the authors use a combination of functional assays, cryoEM structural experiments and mutagenesis to investigate this question. They show that SPT-ORMDL complex co-purify with endogenous ceramide lipids, likely a long chain C24 ceramide, and mutagenesis show that this ceramide binding site plays a key role in suppressing SPT activity. In agreement with past results, the N-terminus of ORMDL3 interacts with SPTLC1 and SPTLC2 and likely interferes with their catalytic activity. Remarkably, the authors show that mutations in the N-terminus destabilize ceramide binding to the inhibitory site and conversely, mutations at the inhibitory binding site destabilize the N-terminus, thus establishing an allosteric mechanism between these two inhibitory regions. Importantly, the authors show that SPLTC1 mutations associated with childhood ALS cause impaired ceramide sensing in the SPT-ORMDL3 complex, providing a mechanistic link between this disease and sphingolipid biosynthesis.

Overall, this is an excellent manuscript. The experiments are well-designed, logically laid out, and appropriately interpreted. This was one of the rare cases when, while reading a manuscript and thinking of the next experiment, I'd satisfactorily find it in the following section. As such, I have no major concerns, only a couple of minor suggestions and a point to perhaps consider in the discussion.

This reviewer fully recognized the significance of our study and only had several minor suggestions that are addressed below.

I realize it is a big ask, but the manuscript would be greatly strengthened if the authors could determine the cryoEM structure of one of the ALS mutants. This would provide a structural framework to interpret their interesting functional results.

We thank this reviewer for the constructive comment. We totally agree with this reviewer that the structure of the ALS mutants would help to interpret the functional results better. Despite multiple attempts, we have been unable to obtain a high-resolution structure of the ALS mutants. Nonetheless, we are confident that the major conclusions drawn from the current study regarding the ceramide sensing mechanism and the implications for impaired ceramide sensing in ALS are robust. We will continue to pursue the cryo-EM study of the ALS mutants in the future as part of ongoing investigations.

The finding that the effects of the mutants is somewhat blunted when assayed in cells compared to when measured in vitro is interesting. I wonder if this could reflect the fact

that in native membranes multiple different kinds of ceramides will be delivered to the complex, whereas in the in vitro experiments only a single type is considered. If the interactions between the SPT-ORMDL3 complex and various ceramide lipids differ slightly depending on chain length and saturation, then it is possible that the mutants might have a different effect in vitro or in cells. If appropriate, this could be added to the discussion.

We appreciate this reviewer for the insightful comment. As suggested by this reviewer, we've added a statement to the revised manuscript (lines 401-405) as follows: "Another possible explanation is that there are numerous ceramide species with diverse chain lengths and saturation levels in native membranes, while only a short chain C6-ceramide was employed for the in vitro assays. The interactions between SPT-ORMDL3 complex and various ceramide species might differ slightly depending on chain lengths and saturation levels, thus causing a different effect on the mutants in cells versus in vitro."

I would suggest the authors explain in a little more detail the rationale for the mutagenesis in Fig. 2. Are the residues considered the only ones with side chains within interaction distance? Was a cut-off distance used in their analysis? if so, this should be stated. For example, in Fig. 2c it appears that I22 (from ORM DL3) is quite far from the ceramide tail whereas F498 (from SPTLC2) seems close. However, the authors only mutated the former and not the latter.

We thank this reviewer for the considerate comment. The distances between I22 or F498 and the ceramide molecule are very similar, bearing values of 5.3 Å or 5 Å, respectively. As I22 possesses a smaller side chain, we suspected that mutating I22 instead of F498 would cause more dramatic effects on the SPT-ORMDL3 complex. For clarification, a statement "For the hydrophobic interface regarding the LCB chain of ceramide, only Phe63 of ORM DL3 was mutated to positively-charged arginine, as the other interface residues in IMH of SPTLC2 are buried in the hydrophobic membrane (Fig. 3b, left). For the hydrophobic interface considering the acyl-chain of ceramide, Val16 and Ile22 of ORM DL3, and Ile503 of SPTLC2 were chosen to be replaced by positively-charged arginine as they possess smaller side chains compared to Phe498 of SPTLC2 (Fig. 3b, right)." was added to the revised manuscript (lines 222-228).

The structural shifts in Fig. 4b are quite difficult to make out. I would suggest that the authors use colors with a bit more contrast than light gray and pale yellow for the two structures, and/or color the ceramide molecule in a color different from the light gray used for the WT SPT-ORMDL3.

Point taken. The colors of WT SPT-ORMDL3 and ceramide were changed to dark gray and pink in Fig. 5b (original Fig. 4b) in the revised manuscript.

We thank the reviewer for his/her time and constructive comments.

REVIEWERS' COMMENTS

Reviewer #2 (Remarks to the Author):

The authors did an excellent job in addressing my previous concerns and submitted a significantly improved version of an already compelling study on how SPT-ORMDL senses ceramides to establish sphingolipid homeostasis. Even though I did not ask for this explicitly, I much appreciate their efforts to utilize bifunctional ceramide analogs to directly test ceramide binding and acknowledge their conclusion that this approach may not be suitable for determining binding kinetics. Their finding that purified SPT-ORMDL3-N13A and SPT-ORMDL3-deltaN2 have a reduced ceramide content in comparison to the wildtype enzyme complex is in line with the structural data and provides additional support for the working model for the homeostatic regulation of SPT-ORMDL by ceramide presented in Fig. 7.

Response to Reviewers:

Reviewer #2:

The authors did an excellent job in addressing my previous concerns and submitted a significantly improved version of an already compelling study on how SPT-ORMDL senses ceramides to establish sphingolipid homeostasis. Even though I did not ask for this explicitly, I much appreciate their efforts to utilize bifunctional ceramide analogs to directly test ceramide binding and acknowledge their conclusion that this approach may not be suitable for determining binding kinetics. Their finding that purified SPT-ORMDL3-N13A and SPT-ORMDL3-deltaN2 have a reduced ceramide content in comparison to the wildtype enzyme complex is in line with the structural data and provides additional support for the working model for the homeostatic regulation of SPT-ORMDL by ceramide presented in Fig. 7.

We thank the reviewer for his/her appreciation of our revisions and thank for his/her time and constructive comments.